# Transformation-Invariant Learning and Theoretical Guarantees for OOD Generalization

**Omar Montasser**
Yale University
omar.montasser@yale.edu

**Han Shao**
Harvard University
han@ttic.edu

**Emmanuel Abbe**
EPFL and Apple
emmanuel.abbe@epfl.ch

## Abstract

Learning with identical train and test distributions has been extensively investigated both practically and theoretically. Much remains to be understood, however, in statistical learning under distribution shifts. This paper focuses on a distribution shift setting where train and test distributions can be related by classes of (data) transformation maps. We initiate a theoretical study for this framework, investigating learning scenarios where the target class of transformations is either known or unknown. We establish learning rules and algorithmic reductions to Empirical Risk Minimization (ERM), accompanied with learning guarantees. We obtain upper bounds on the sample complexity in terms of the VC dimension of the class composing predictors with transformations, which we show in many cases is not much larger than the VC dimension of the class of predictors. We highlight that the learning rules we derive offer a game-theoretic viewpoint on distribution shift: a learner searching for predictors and an adversary searching for transformation maps to respectively minimize and maximize the worst-case loss.

## 1 Introduction

It is desirable to train machine learning predictors that are robust to distribution shifts. In particular when data distributions vary based on the environment, or when part of the domain is not sampled at training such as in reasoning tasks. How can we train predictors that generalize beyond the distribution from which the training examples are drawn from? A common challenge that arises when tackling out-of-distribution generalization is capturing the structure of distribution shifts. A common approach is to mathematically describe such shifts through distance or divergence measures, as in prior work on domain adaptation theory [e.g., Redko et al., 2020] and distributionally robust optimization [e.g., Duchi and Namkoong, 2021].

In this paper, we put forward a new formulation for out-of-distribution generalization. Our formulation offers a conceptually different perspective from prior work, where we describe the structure of distribution shifts through data transformations. We consider an unknown distribution $\mathcal{D}$ over $\mathcal{X} \times \mathcal{Y}$ which can be thought of as the "training" or "source" distribution from which training examples are drawn, and a collection of data transformation maps $\mathcal{T} = \{T : \mathcal{X} \to \mathcal{X}\}$ which can be thought of as encoding "target" distribution shifts, hence denoted as $\{T(\mathcal{D})\}_{T \in \mathcal{T}}$. We consider a covariate shift setting where labels are *not* altered or changed under transformations $T \in \mathcal{T}$, and we write $T(\mathcal{D})$ for notational convenience. Our goal, which will be formalized further shortly, is to learn a single predictor $\hat{h}$ that performs well *uniformly* across *all* distributions $\{T(\mathcal{D})\}_{T \in \mathcal{T}}$.

We view this formulation as enabling a different way to describe distribution shifts through transformations $\mathcal{T} = \{T : \mathcal{X} \to \mathcal{X}\}$. The collection of transformations $\mathcal{T}$ can be viewed as either: (a) given to the learning algorithm as part of the problem, or (b) chosen by the learning algorithm. View (a) represents scenarios where the target distribution shifts are known and specified by some downstream application (e.g., learning a classifier that is invariant to image rotations and translations). View (b)

represents scenarios where there is uncertainty or there are no pre-specified target distribution shifts and we would like to perform maximally well relative to an expressive collection of transformations. We highlight next several problems of interest that can be captured by this formulation. We refer the reader to Section 7 for a more detailed discussion in the context of prior work.

- Covariate Shift & Domain Adaptation. By Brenier's theorem [Brenier, 1991], when $\mathcal{X} = \mathbb{R}^d$, then under mild assumptions, for any source distribution $P$ over $\mathcal{X}$ and target distribution $Q$ over $\mathcal{X}$, there exists a transformation $T : \mathcal{X} \to \mathcal{X}$ such that $Q = T(P)$. Thus, by choosing an expressive collection of transformations $\mathcal{T}$, we can address arbitrary covariate shifts.

- Transformation-Invariant Learning. In many applications, it is desirable to train predictors that are invariant to transformations or data preprocessing procedures representing different "environments" (e.g., an image classifier deployed in different hospitals, or a self-driving car operating in different cities).

- Representative Sampling. In many applications, there may be challenges in collecting "representative" training data. For instance, in learning Logic or Arithmetic tasks [Abbe et al., 2023], the combinatorial nature of the data makes it not possible to cover well all parts of the domain. E.g., there is always a limit to the length of the problem considered at training, or features may not be homogeneously represented at training (bias towards certain digits etc.). Choosing a suitable collection of transformations $\mathcal{T}$ under which the target function is invariant can help to model in such cases.

- Adversarial Attacks. Test-time adversarial attacks such as adversarial patches in vision tasks [Brown et al., 2017, Karmon et al., 2018], attack prompts in large language models [Zou et al., 2023], and "universal attacks" [Moosavi-Dezfooli et al., 2017] can all be viewed as instantiations constructing specific transformations $\mathcal{T}$.

**Our Contributions.** Let $\mathcal{X}$ be the instance space and $\mathcal{Y} = \{\pm 1\}$ the label space. Let $\mathcal{H} \subseteq \mathcal{Y}^{\mathcal{X}}$ be a hypothesis class, and denote by $\mathrm{vc}(\mathcal{H})$ its VC dimension. Consider a collection of transformations $\mathcal{T} = \{T : \mathcal{X} \to \mathcal{X}\}$, and some unknown distribution $\mathcal{D}$ over $\mathcal{X} \times \mathcal{Y}$. Let $\mathrm{err}(h, T(\mathcal{D})) = \Pr_{(x,y) \sim \mathcal{D}} [h(T(x)) \neq y]$ be the error of predictor $h$ on transformed distribution $T(\mathcal{D})$.

Given a training sample $S = \{(x_1, y_1), \ldots, (x_m, y_m)\} \sim \mathcal{D}^m$, we are interested in learning a predictor $\hat{h}$ with *uniformly small risk* across all transformations $T \in \mathcal{T}$. Formally,

$$\sup_{T \in \mathcal{T}} \mathrm{err}(\hat{h}, T(\mathcal{D})) \leq \mathsf{OPT}_\infty + \varepsilon, \text{ where } \mathsf{OPT}_\infty := \inf_{h^\star \in \mathcal{H}} \sup_{T \in \mathcal{T}} \{\mathrm{err}(h^\star, T(\mathcal{D}))\}. \quad (1)$$

This objective is similar to that considered in prior work on distributionally robust optimization [Duchi and Namkoong, 2021] and multi-distribution learning [Haghtalab et al., 2022]. The main difference is that in this work we are describing the collection of "target" distributions $\{T(\mathcal{D})\}_{T \in \mathcal{T}}$ as transformations of the "source" distribution $\mathcal{D}$. This allows us to obtain new upper bounds on the sample complexity of learning under distribution shifts based on the VC dimension of the composition of $\mathcal{H}$ with $\mathcal{T}$, denoted $\mathrm{vc}(\mathcal{H} \circ \mathcal{T})$ (see Equation (3)). We describe next our results (informally):

1. In Section 2 (Theorem 2.1), we show that, given the knowledge of any hypothesis class $\mathcal{H}$ and any collection of transformations $\mathcal{T}$, by minimizing the empirical worst case risk, we can solve Objective 1 with sample complexity bounded by $\mathrm{vc}(\mathcal{H} \circ \mathcal{T})$. Furthermore, in Theorem 2.2, we show that the sample complexity of any *proper* learning rule is bounded from below by $\Omega(\mathrm{vc}(\mathcal{H} \circ \mathcal{T}))$.

2. In Section 3 (Theorem 3.1), we consider a more challenging scenario in which $\mathcal{H}$ is unknown. Instead, we are only given an ERM oracle for $\mathcal{H}$. We then present a generic algorithmic reduction (Algorithm 1) solving Objective 1 using only an ERM oracle for $\mathcal{H}$, when the collection $\mathcal{T}$ is finite. This is established by solving a zero-sum game where the $\mathcal{H}$-player runs ERM and the $\mathcal{T}$-player runs Multiplicative Weights [Freund and Schapire, 1997].

3. In Section 4 (Theorem 4.1), we consider situations where we *do not know* which transformations are relevant (or important) for the learning task at hand, and so we pick an expressive collection $\mathcal{T}$ and aim to perform well on as many transformations as possible. We then present a different generic learning rule (Equation (4)) that learns a predictor $\hat{h}$ achieving low error (say $\varepsilon$) on as many target distributions in $\{T(\mathcal{D})\}_{T \in \mathcal{T}}$ as possible.

4. In Section 5 (Theorems 5.1 & E.1), we extend our learning guarantees to a slightly different objective, Objective 7, that can be favorable to Objective 1 when there is heterogeneity in the noise

across different transformations. This is inspired by Agarwal and Zhang [2022] who introduced this objective.

## 2 Minimizing Worst-Case Risk

If we have access to, or know, the hypothesis class $\mathcal{H}$ and the collection of transformations $\mathcal{T}$, then the most direct and intuitive way of solving Objective 1 is minimizing the empirical worst-case risk. Specifically,

$$\hat{h} \in \underset{h \in \mathcal{H}}{\operatorname{argmin}} \max_{T \in \mathcal{T}} \left\{ \frac{1}{m} \sum_{i=1}^{m} \mathbb{1}\left[h(T(x_i)) \neq y_i\right] \right\}. \tag{2}$$

We highlight that this learning rule offers a game-theoretic perspective on distribution shift, where the $\mathcal{H}$-player searches for a predictor $h \in \mathcal{H}$ to minimize the worst-case error while the $\mathcal{T}$-player searches for a transformation $T \in \mathcal{T}$ to maximize the worst-case error. For instance, both predictors $\mathcal{H}$ and transformations $\mathcal{T}$ can be parameterized by neural network architectures, which is an interesting direction to explore further. We note that similar min-max optimization problems have appeared before in the literature on adversarial examples and generative adversarial networks [e.g., Madry et al., 2018, Goodfellow et al., 2020].

We present next a PAC-style learning guarantee for this learning rule which offers the interpretation that solving the min-max optimization problem in Equation (1) yields a predictor $\hat{h} \in \mathcal{H}$ that generalizes to the collection of transformations $\mathcal{T}$. We show that the sample complexity of this learning rule is bounded by the VC dimension of the composition of $\mathcal{H}$ with $\mathcal{T}$, where

$$\mathcal{H} \circ \mathcal{T} := \{h \circ T : h \in \mathcal{H}, T \in \mathcal{T}\}, \text{ where } (h \circ T)(x) = h(T(x)) \ \forall x \in \mathcal{X}. \tag{3}$$

**Theorem 2.1.** *For any class $\mathcal{H}$, any collection of transformations $\mathcal{T}$, any $\varepsilon, \delta \in (0, 1/2)$, any distribution $\mathcal{D}$, with probability at least $1 - \delta$ over $S \sim \mathcal{D}^{m(\varepsilon,\delta)}$ where $m(\varepsilon, \delta) = O\left(\frac{\operatorname{vc}(\mathcal{H} \circ \mathcal{T}) + \log(1/\delta)}{\varepsilon^2}\right)$,*

$$\sup_{T \in \mathcal{T}} \operatorname{err}(\hat{h}, T(D)) \leq \mathsf{OPT}_\infty + \varepsilon.$$

*Proof.* The proof follows from invoking uniform convergence guarantees with respect to the composition $\mathcal{H} \circ \mathcal{T}$ (see Proposition A.1 in Appendix A) and the definition of $\hat{h}$ described in Equation (2). Let $h^\star \in \mathcal{H}$ be an a-priori fixed predictor (independent of sample $S$) attaining $\mathsf{OPT}_\infty = \inf_{h \in \mathcal{H}} \sup_{T \in \mathcal{T}} \operatorname{err}(h, T(\mathcal{D}))$ (or $\varepsilon$-close to it). By setting $m(\varepsilon, \delta) = O\left(\frac{\operatorname{vc}(\mathcal{H} \circ \mathcal{T}) + \log(1/\delta)}{\varepsilon^2}\right)$ and invoking Proposition A.1, we have the guarantee that with probability at least $1 - \delta$ over $S \sim \mathcal{D}^{m(\varepsilon,\delta)}$,

$$(\forall h \in \mathcal{H}) \, (\forall T \in \mathcal{T}) : |\operatorname{err}(h, T(S)) - \operatorname{err}(h, T(\mathcal{D}))| \leq \varepsilon.$$

Since $\hat{h}, h^\star \in \mathcal{H}$, the inequality above implies that

$$\forall T \in \mathcal{T} : \ \operatorname{err}(\hat{h}, T(\mathcal{D})) \leq \operatorname{err}(\hat{h}, T(S)) + \varepsilon.$$
$$\forall T \in \mathcal{T} : \ \operatorname{err}(h^\star, T(S)) \leq \operatorname{err}(h^\star, T(\mathcal{D})) + \varepsilon.$$

Furthermore, by definition, since $\hat{h}$ minimizes the empirical objective, it holds that

$$\sup_{T \in \mathcal{T}} \operatorname{err}(\hat{h}, T(S)) \leq \sup_{T \in \mathcal{T}} \operatorname{err}(h^\star, T(S)).$$

By combining the above, we get

$$\sup_{T \in \mathcal{T}} \operatorname{err}(\hat{h}, T(\mathcal{D})) \leq \sup_{T \in \mathcal{T}} \operatorname{err}(\hat{h}, T(S)) + \varepsilon \leq \sup_{T \in \mathcal{T}} \operatorname{err}(h^\star, T(S)) + \varepsilon \leq \mathsf{OPT}_\infty + 2\varepsilon. \qquad \square$$

We show next that $\operatorname{vc}(\mathcal{H} \circ \mathcal{T})$ can be much higher than $\operatorname{vc}(\mathcal{H})$ and the dependency on $\operatorname{vc}(\mathcal{H} \circ \mathcal{T})$ is tight for all proper learning rules, which includes the learning rule described in Equation (2) and more generally any learning rule that is restricted to outputting a classifier in $\mathcal{H}$.

**Theorem 2.2.** *$\forall k \in \mathbb{N}$, $\exists \mathcal{X}, \mathcal{H}, \mathcal{T}$ such that $\operatorname{vc}(\mathcal{H}) = 1$ but $\operatorname{vc}(\mathcal{H} \circ \mathcal{T}) \geq k$, and the sample complexity of* any proper *learning rule $\mathbb{A} : (\mathcal{X} \times \mathcal{Y})^* \to \mathcal{H}$ solving Objective 1 is* at least $\Omega(\operatorname{vc}(\mathcal{H} \circ \mathcal{T}))$.

A proof is deferred to Appendix C. We remark that the sample complexity cannot be improved by proper learning rules and this leaves open the possibility of improving the sample complexity with *improper* learning rules. There are many examples in the literature where there are sample complexity gaps between proper and improper learning [e.g., Angluin, 1987, Daniely and Shalev-Shwartz, 2014, Foster et al., 2018, Montasser et al., 2019, Alon et al., 2021]. In particular, it appears that we encounter in this work a phenomena similar to what occurs in adversarially robust learning [Montasser et al., 2019]. Nonetheless, even at the expense of (potentially) higher sample complexity, we believe that there is value in the simplicity of the learning rule described in Equation (2), and exploring ways of implementing it is an interesting direction beyond the scope of this work.

### 2.1 Examples and Instantiations of Guarantees

To demonstrate the utility of our generic result in Theorem 2.1, we discuss next a few general cases where we can bound the VC dimension of $\mathcal{H}$ composed with $\mathcal{T}$, $\mathrm{vc}(\mathcal{H} \circ \mathcal{T})$. This allows us to obtain new learning guarantees with respect to classes of distribution shifts that are *not* covered by prior work, to the best of our knowledge.

**Linear Transformations.** Consider $\mathcal{T}$ being a (potentially infinite) collection of *linear* transformations. For example, in vision tasks, this includes many transformations that have been widely studied in practice such as rotations, translations, maskings, adding random noise (or any fixed a-priori arbitrary noise), and their compositions [Engstrom et al., 2019, Hendrycks and Dietterich, 2019].

Interestingly, for a broad range of hypothesis classes $\mathcal{H}$, we can show that $\mathrm{vc}(\mathcal{H} \circ \mathcal{T}) \leq \mathrm{vc}(\mathcal{H})$ without incurring any dependence on the complexity of $\mathcal{T}$. Specifically, the result applies to any function class $\mathcal{H}$ that consists of a linear mapping followed by an arbitrary mapping. This includes feed-forward neural networks with any activation function, and modern neural network architectures (e.g., CNNs, ResNets, Transformers). We find the implication of this bound to be interesting, because it suggests (along with Theorem 2.1) that the learning rule in Equation (2) can generalize to linear transformations with sample complexity that is not greater than the sample complexity of standard PAC learning. We formally present the lemma below, and defer the proof to Appendix B.

**Lemma 2.3.** *For any collection of* linear *transformations $\mathcal{T}$ and any hypothesis class of the form* $\mathcal{H} = \left\{ f \circ W : \mathbb{R}^d \to \mathcal{Y} \mid W \in \mathbb{R}^{k \times d} \wedge f : \mathbb{R}^k \to \mathcal{Y} \right\}$, *it holds that* $\mathrm{vc}(\mathcal{H} \circ \mathcal{T}) \leq \mathrm{vc}(\mathcal{H})$.

**Non-Linear Transformations.** Consider $\mathcal{T}$ being a (potentially infinite) collection of *non-linear* transformations parameterized by a feed-forward neural network architecture, where each $T = W_L \circ \phi \circ \cdots \phi \circ W_2 \circ \phi \circ W_1$ and $\phi(\cdot) = \max \{0, \cdot\}$ is the ReLU activation function. Similarly, consider a hypothesis class $\mathcal{H}$ that is parameterized by a (different) feed-forward neural network architecture, where each $h = \mathrm{sign} \circ \tilde{W}_H \circ \phi \circ \cdots \phi \circ \tilde{W}_2 \circ \phi \circ \tilde{W}_1$. Observe that the composition $\mathcal{H} \circ \mathcal{T}$ consists of (deeper) feed-forward neural networks, where $h \circ T = \mathrm{sign} \circ \tilde{W}_H \circ \phi \circ \cdots \phi \circ \tilde{W}_2 \circ \phi \circ \tilde{W}_1 \circ W_L \circ \phi \circ \cdots \phi \circ W_2 \circ \phi \circ W_1$. Thus, we can bound $\mathrm{vc}(\mathcal{H} \circ \mathcal{T})$ by appealing to classical results bounding the VC dimension of feed-forward neural networks. For example, according to Bartlett et al. [2019], it holds that $\mathrm{vc}(\mathcal{H} \circ \mathcal{T}) \leq O\left((H + L)P_{\mathcal{H} \circ \mathcal{T}} \log(P_{\mathcal{H} \circ \mathcal{T}})\right)$, where $H + L$ is the depth of the networks in $\mathcal{H} \circ \mathcal{T}$ and $P_{\mathcal{H} \circ \mathcal{T}}$ is the number of parameters of the networks in $\mathcal{H} \circ \mathcal{T}$ (which is $P_{\mathcal{H}} + P_{\mathcal{T}}$). In this context, Theorem 2.1 and Equation (2) present a new learning guarantee against distribution shifts parameterized by non-linear transformations induced with feed-forward neural networks.

**Transformations on the Boolean hypercube.** The Boolean hypercube has also received attention recently as a case-study for distribution shifts in Logic or Arithmetic tasks[Abbe et al., 2023]. We show next that when the instance space $\mathcal{X} = \{0, 1\}^d$, we can bound the VC dimension of $\mathcal{H} \circ \mathcal{T}$ from above by the sum of the VC dimension of $\mathcal{H}$ and the VC dimensions of $\{\mathcal{T}_i\}_{i=1}^{d}$ where each $\mathcal{T}_i = \{x \mapsto T(x)_i : T \in \mathcal{T}\}$ is a function class resulting from restricting transformations $T : \{0, 1\}^d \to \{0, 1\}^d \in \mathcal{T}$ to output only the $i^{\mathrm{th}}$ bit. The proof is deferred to Appendix B

**Lemma 2.4.** *When $\mathcal{X} = \{0, 1\}^d$, for any hypothesis class $\mathcal{H}$ and any collection of transformations* $\mathcal{T}$, $\mathrm{vc}(\mathcal{H} \circ \mathcal{T}) \leq O(\log d)(\mathrm{vc}(\mathcal{H}) + \sum_{i=1}^{d} \mathrm{vc}(\mathcal{T}_i))$, *where each $\mathcal{T}_i = \{x \mapsto T(x)_i : T \in \mathcal{T}\}$.*

In this context, Theorem 2.1 and Equation (2) present a new learning guarantee against arbitrary distribution shifts parameterized by transformations on the Boolean hypercube, where the sample complexity (potentially) grows with the complexity of the transformations as measured by the VC

dimension. We note however that this learning guarantee does not address the problem of length generalization, since we restrict to transformations that preserve domain length.

**Adversarial Attacks.** In adversarially robust learning, a test-time attacker is typically modeled as a pertubation function $\mathcal{U} : \mathcal{X} \to 2^{\mathcal{X}}$, which specifies for each test-time example $x$ a set of possible adversarial attacks $\mathcal{U}(x) \subseteq \mathcal{X}$ that the attacker can choose from at test-time [Montasser et al., 2019]. The robust risk of a predictor $\hat{h}$ is then defined as: $\mathbb{E}_{(x,y)\sim\mathcal{D}} \left[ \sup_{z\in\mathcal{U}(x)} \mathbb{1}\left[ \hat{h}(z) \neq y \right] \right]$. On the other hand, the framework we consider in this paper can be viewed as restricting a test-time attacker to commit to a set of attacks $\mathcal{T} = \{T : \mathcal{X} \to \mathcal{X}\}$ without knowledge of the test-time samples, and the risk of a predictor $\hat{h}$ is then defined as: $\sup_{T\in\mathcal{T}} \mathbb{E}_{(x,y)\sim\mathcal{D}} \mathbb{1}\left[ \hat{h}(T(x)) \neq y \right]$. While less general, our framework still captures several interesting adversarial attacks in practice which are constructed before seeing test-time examples, such as adversarial patches in vision tasks [Brown et al., 2017, Karmon et al., 2018] and attack prompts for large language models [Zou et al., 2023] can be represented with linear transformations.

# 3 Unknown Hypothesis Class: Algorithmic Reductions to ERM

Implementing the learning rule in Equation (2) crucially requires knowing the base hypothesis class $\mathcal{H}$ and the transformations $\mathcal{T}$, which may not be feasible in many scenarios. Moreover, in many applications we only have black-box access to an off-the-shelve supervised learning method such as an ERM for $\mathcal{H}$. Hence, in this section, we study the following question:

Can we solve Objective 1 using only an ERM oracle for $\mathcal{H}$?

We prove yes, and we present next generic oracle-efficient reductions solving Objective 1 using only an ERM oracle for $\mathcal{H}$. We consider two cases,

**Realizable Case.** When $\mathsf{OPT}_\infty = 0$, i.e., $\exists h^\star \in \mathcal{H}$ such that $\forall T \in \mathcal{T} : \mathrm{err}(h^\star, T(\mathcal{D})) = 0$, there is a simple reduction to solve Objective 1 using a *single call* to an ERM oracle for $\mathcal{H}$. The idea is to inflate the training dataset $S$ to include all possible transformations $\mathcal{T}(S) = \{(T(x), y) : (x, y) \in S \land T \in \mathcal{T}\}$ (similar to data augmentation), and then run ERM on $\mathcal{T}(S)$. Formal guarantee and proof are deferred to Appendix D. It is also possible, via a fairly standard boosting argument, to achieve a similar learning guarantee using multiple ERM calls (specifically, $O(\log |\mathcal{T}(S)|) \leq O(\log(|S| |\mathcal{T}|))$), where each ERM call is on a sample of size $O(\mathrm{vc}(\mathcal{H}))$. So, we get a tradeoff between the size of a dataset given to ERM on a single call, and the total number of calls to ERM.

**Agnostic Case.** When $\mathsf{OPT}_\infty > 0$, the simple reduction above no longer works. Specifically, the issue is that running a single ERM on the inflation $\mathcal{T}(S)$ effectively minimizes average error over transformations $T \in \mathcal{T}$ as opposed to minimizing maximum error over transformations $T \in \mathcal{T}$. So, $\mathsf{OPT}_\infty > 0$, by definition, implies there is no predictor $h \in \mathcal{H}$ that is consistent (i.e., zero error) on every transformation $T(S), T \in \mathcal{T}$, thus minimizing average error over transformations can be bad.

To overcome this limitation, we present a different reduction (Algorithm 1) that minimizes Objective 1 by solving a zero-sum game where the $\mathcal{H}$-player runs ERM and the $\mathcal{T}$-player runs Multiplicative Weights [Freund and Schapire, 1997]. This can be viewed as solving a sequence of weighted-ERM problems (with weights over transformations), where Multiplicative Weights determines the weight of each transformation.

**Theorem 3.1.** *For any class $\mathcal{H}$, collection of transformations $\mathcal{T}$, distribution $\mathcal{D}$ and any $\varepsilon, \delta \in (0, 1/2)$, with probability at least $1 - \delta$ over $S \sim \mathcal{D}^{m(\varepsilon,\delta)}$, where $m(\varepsilon,\delta) \leq O(\frac{\mathrm{vc}(\mathcal{H}\circ\mathcal{T})+\log(1/\delta)}{\varepsilon^2})$, running Algorithm 1 on $S$ for $R \geq \frac{8\ln|\mathcal{T}|}{\varepsilon^2}$ rounds produces $\bar{h} = \frac{1}{R}\sum_{r=1}^{R} h_r$ satisfying*

$$\forall T \in \mathcal{T} : \Pr_{\substack{(x,y)\sim D \\ r\sim\mathrm{Unif}\{1,\ldots,R\}}} [h_r(T(x)) \neq y] \leq \mathsf{OPT}_\infty + \varepsilon.$$

*Remark* 3.2. When $\mathcal{T}$ is a *finite* collection of transformations, we can bound $\mathrm{vc}(\mathcal{H} \circ \mathcal{T})$ from above by $O(\mathrm{vc}(\mathcal{H}) + \log |\mathcal{T}|)$ using the Sauer-Shelah-Perels Lemma [Sauer, 1972]. See Lemma B.1 and proof in Appendix B.

---

**Algorithm 1:** Reduction to Minimize Worst-Case Risk

---

**Input:** Black-box $\text{ERM}_{\mathcal{H}}$, dataset $S = \{(x_1, y_1), \ldots, (x_m, y_m)\}$, and transformations $\mathcal{T}$.

1 For each $T \in \mathcal{T}$, set $Q_1(T) = \frac{1}{|\mathcal{T}|}$.

2 Set number of rounds $R = \frac{8 \ln |\mathcal{T}|}{\varepsilon^2}$.

3 **for** $1 \leq r \leq R$ **do**

4    Run $\text{ERM}_{\mathcal{H}}$ on $m_{\text{ERM}}$ i.i.d. samples drawn from the distribution induced by $Q_r$ over $\mathcal{T}$ and $\text{Unif}(S)$, and let $h_r$ denote its output.

5    For each $T \in \mathcal{T}$, update $Q_{r+1}(T) = \frac{Q_r(T) \exp(-\eta(1 - \text{err}(h_r, T(S))))}{Z_r}$, where $Z_r$ is a normalization factor such that $Q_{r+1}$ is a distribution.

**Output:** $\frac{1}{R} \sum_{t=1}^{R} h_r$.

---

*Proof of Theorem 3.1.* Let $S = \{(x_1, y_1), \ldots, (x_m, y_m)\}$ be an arbitrary dataset. By setting $R \geq \frac{8 \ln |\mathcal{T}|}{\varepsilon^2}$ and invoking Lemma D.2, which is a helpful lemma (statement and proof in Appendix D) that instantiates the regret guarantee of Multiplicative Weights in our context, we are guaranteed that Algorithm 1 produces a sequence of distributions $Q_1, \ldots, Q_R$ over $\mathcal{T}$ that satisfy

$$\max_{T \in \mathcal{T}} \frac{1}{R} \sum_{r=1}^{R} \text{err}(h_r, T(S)) \leq \frac{1}{R} \sum_{r=1}^{R} \mathop{\mathbb{E}}_{T \sim Q_r} \text{err}(h_r, T(S)) + \frac{\varepsilon}{4}.$$

At each round $r$, observe that Step 4 in Algorithm 1 draws an i.i.d. sample from a distribution $P_r$ over $\mathcal{X} \times \mathcal{Y}$ that is defined by $Q_r$ over $\mathcal{T}$ and $\text{Unif}(S)$, and since $\text{ERM}_{\mathcal{H}}$ is an $(\varepsilon, \delta)$-agnostic-PAC-learner for $\mathcal{H}$, Step 5 guarantees that

$$\mathop{\mathbb{E}}_{T \sim Q_r} \text{err}(h_r, T(S)) = \mathop{\mathbb{E}}_{T \sim Q_r} \frac{1}{m} \sum_{i=1}^{m} \mathbb{1}\{h_r(T(x_i)) \neq y_i\} \leq \min_{h \in \mathcal{H}} \mathop{\mathbb{E}}_{T \sim Q_r} \text{err}(h, T(S)) + \frac{\varepsilon}{4} \leq \min_{h^\star \in \mathcal{H}} \max_{T \in \mathcal{T}} \text{err}(h^\star, T(S)) + \frac{\varepsilon}{4}.$$

Combining the above inequalities implies that

$$\max_{T \in \mathcal{T}} \frac{1}{R} \sum_{r=1}^{R} \text{err}(h_r, T(S)) \leq \min_{h^\star \in \mathcal{H}} \max_{T \in \mathcal{T}} \text{err}(h^\star, T(S)) + \frac{\varepsilon}{2}.$$

Finally, by appealing to uniform convergence over $\mathcal{H} \circ \mathcal{T}$ (Proposition A.1), with probability at least $1 - \delta$ over $S \sim \mathcal{D}^m$,

$$\max_{T \in \mathcal{T}} \frac{1}{R} \sum_{r=1}^{R} \text{err}(h_r, T(\mathcal{D})) \leq \max_{T \in \mathcal{T}} \frac{1}{R} \sum_{r=1}^{R} \text{err}(h_r, T(S)) + \frac{\varepsilon}{4} \leq \min_{h^\star \in \mathcal{H}} \max_{T \in \mathcal{T}} \text{err}(h^\star, T(S)) + \frac{\varepsilon}{2} + \frac{\varepsilon}{4}$$

$$\leq \min_{h^\star \in \mathcal{H}} \max_{T \in \mathcal{T}} \text{err}(h^\star, T(\mathcal{D})) + \frac{\varepsilon}{2} + 2\frac{\varepsilon}{4} = \text{OPT}_\infty + \varepsilon. \qquad \square$$

**On finiteness of $\mathcal{T}$.** We argue informally that requiring $\mathcal{T}$ to be finite is necessary in general when only an $\text{ERM}$ oracle for $\mathcal{H}$ is allowed. For example, consider a distribution supported on a single point $(x, -)$ on the real line where $x = 5$, and transformations $T_i(x) = x + i$ for all $i \geq 1$ induced by some collection $\{T_i\}_{i \in \mathbb{N}}$. Calling ERM on a finite subset of these transformations $T_{i_1}, \ldots, T_{i_k}$ could return a predictor that labels $x, x + i_1, x + i_2, \ldots, x + i_k$ with a label $-$ and labels $x + i_k + 1, \ldots$ with $+$ (e.g., if $\mathcal{H}$ is thresholds) which fails to satisfy Objective 1. But it would be interesting to explore additional structural conditions that would enable handling infinite $\mathcal{T}$, and leave this to future work.

## 4  Unknown Invariant Transformations

When we have a large collection of transformations $\mathcal{T}$ and there is uncertainty about which transformations $T \in \mathcal{T}$ under-which we can simultaneously achieve low error using a base class $\mathcal{H}$, the learning rule presented in Section 2 (Equation 1) can perform badly. We illustrate this with the following concrete example:

*Example* 1. Consider a class $\mathcal{H} = \{h_1, h_2, h_3\}$, a collection of transformations $\mathcal{T} = \{T_1, T_2, T_3\}$, and a distribution $\mathcal{D}$ with risks (errors) as reported in the table.

|       | $T_1(\mathcal{D})$ | $T_2(\mathcal{D})$ | $T_3(\mathcal{D})$ |
|-------|--------|--------|--------|
| $h_1$ | 1%     | 1%     | 49%    |
| $h_2$ | 1%     | 49%    | 49%    |
| $h_3$ | 49%    | 49%    | 49%    |

Then, solving Objective 1 may return predictor $h_3$ where $\forall T \in \mathcal{T} : \text{err}(h_3, T(\mathcal{D})) = 49\%$, since we only need to compete with the worst-case risk $\text{OPT}_\infty = 49\%$. However, predictor $h_1$ is arguably better since it achieves a low error of $1\%$ on at least two out of the three transformations.

To address this limitation, we switch to a different learning goal—achieving low error under as many transformations as possible. We present next a different generic learning rule for any class $\mathcal{H}$ and any collection of transformations $\mathcal{T}$, that enjoys a different guarantee from the learning rule presented in Section 2. In particular, it can be thought of as greedy since it maximizes the number of transformations under which low empirical error is possible, but also conservative since it ignores transformations under which low empirical error is not possible. Specifically, given a training dataset $S$, the learning rule searches for a predictor $\hat{h} \in \mathcal{H}$ that achieves low empirical error on as many transformations $T \in \mathcal{T}$ as possible, say $\text{err}(\hat{h}, T(S)) \leq \varepsilon$.

$$\hat{h} \in \underset{h \in \mathcal{H}}{\arg\max} \sum_{T \in \mathcal{T}} \mathbb{1}\left[\text{err}(h, T(S)) \leq \varepsilon\right]. \tag{4}$$

Another way of thinking about this learning rule is that it provides us with more flexibility in choosing the collection of transformations $\mathcal{T}$, since the learning rule is not stringent on achieving low error on all transformations but instead attempts to achieve low error on as many transformations as allowed by the base class $\mathcal{H}$. Thus, this is useful in situations where there is uncertainty in choosing the collection of transformations. We present next the formal learning guarantee for this learning rule,

**Theorem 4.1.** *For any class $\mathcal{H}$, any* countable *collection of transformations $\mathcal{T}$, any distribution $\mathcal{D}$ and any $\varepsilon, \delta \in (0,1)$, with probability at least $1 - \delta$ over $S \sim \mathcal{D}^m$, where $m = O\left(\frac{\text{vc}(\mathcal{H} \circ \mathcal{T}) \log(1/\varepsilon) + \log(1/\delta)}{\varepsilon}\right)$, then*

$$\sum_{T \in \mathcal{T}} \mathbb{1}\left[\text{err}(\hat{h}, T(\mathcal{D})) \leq 3\varepsilon\right] \geq \max_{h^\star \in \mathcal{H}} \sum_{T \in \mathcal{T}} \mathbb{1}\left[\text{err}(h^\star, T(\mathcal{D})) \leq \frac{\varepsilon}{3}\right].$$

*Furthermore, it holds that $\forall T \in \mathcal{T}$: $\text{err}(\hat{h}, T(\mathcal{D})) \leq \text{err}(\hat{h}, T(S)) + \sqrt{\text{err}(\hat{h}, T(S))\frac{\varepsilon}{3}} + \frac{\varepsilon}{3}$.*

*Remark* 4.2. We can generalize the result above to any prior over the transformations $\mathcal{T}$, encoded as a weight function $w : \mathcal{T} \to [0,1]$ such that $\sum_{T \in \mathcal{T}} w(T) \leq 1$. By maximizing the weighted version of Equation (4) according to $w$, it holds that $\sum_{T \in \mathcal{T}} w(T) \mathbb{1}_{\left[\text{err}(\hat{h}, T(\mathcal{D})) \leq \varepsilon/3\right]} \geq \max_{h^\star \in \mathcal{H}} \sum_{T \in \mathcal{T}} w(T) \mathbb{1}_{\left[\text{err}(h^\star, T(\mathcal{D})) \leq 3\varepsilon\right]}$ with high probability.

*Proof.* The proof follows from the definition of $\hat{h}$ and using optimistic generalization bounds (Proposition A.2). By setting $m(\varepsilon, \delta) = O\left(\frac{\text{vc}(\mathcal{H} \circ \mathcal{T}) \log(1/\varepsilon) + \log(1/\delta)}{\varepsilon}\right)$ and invoking Proposition A.2, we have the guarantee that with probability at least $1 - \delta$ over $S \sim \mathcal{D}^{m(\varepsilon, \delta)}$, $(\forall h \in \mathcal{H})(\forall T \in \mathcal{T})$:

$$\text{err}(h, T(\mathcal{D})) \leq \text{err}(h, T(S)) + \sqrt{\text{err}(h, T(S))\frac{\varepsilon}{3}} + \frac{\varepsilon}{3}, \tag{5}$$

$$\text{err}(h, T(S)) \leq \text{err}(h, T(\mathcal{D})) + \sqrt{\text{err}(h, T(\mathcal{D}))\frac{\varepsilon}{3}} + \frac{\varepsilon}{3}. \tag{6}$$

Since $\hat{h} \in \mathcal{H}$, inequality (4) above implies that $\forall T \in \mathcal{T}$ if $\text{err}(\hat{h}, T(S)) \leq \varepsilon$ then $\text{err}(\hat{h}, T(\mathcal{D})) \leq 3\varepsilon$. Thus,

$$\sum_{T \in \mathcal{T}} \mathbb{1}\left[\text{err}(\hat{h}, T(\mathcal{D})) \leq 3\varepsilon\right] \geq \sum_{T \in \mathcal{T}} \mathbb{1}\left[\text{err}(\hat{h}, T(S)) \leq \varepsilon\right].$$

Furthermore, by definition, since $\hat{h}$ maximizes the empirical objective, it holds that

$$\sum_{T \in \mathcal{T}} \mathbb{1}\left[\text{err}(\hat{h}, T(S)) \leq \varepsilon\right] \geq \sum_{T \in \mathcal{T}} \mathbb{1}\left[\text{err}(h^\star, T(S)) \leq \varepsilon\right].$$

Since $h^\star \in \mathcal{H}$, inequality (5) above implies that $\forall T \in \mathcal{T}$ if $\text{err}(h^\star, T(\mathcal{D})) \le \varepsilon/3$ then $\text{err}(h^\star, T(S)) \le \varepsilon$. Thus,

$$\sum_{T \in \mathcal{T}} \mathbb{1}\left[\text{err}(h^\star, T(S)) \le \varepsilon\right] \ge \sum_{T \in \mathcal{T}} \mathbb{1}\left[\text{err}(h^\star, T(\mathcal{D})) \le \varepsilon/3\right].$$

By combining the above three inequalities,

$$\sum_{T \in \mathcal{T}} \mathbb{1}\left[\text{err}(\hat{h}, T(\mathcal{D})) \le 3\varepsilon\right] \ge \sum_{T \in \mathcal{T}} \mathbb{1}\left[\text{err}(h^\star, T(\mathcal{D})) \le \varepsilon/3\right]. \qquad \square$$

## 5 Extension to Minimizing Worst-Case Regret

When there is heterogeneity in the noise across the different distributions, Agarwal and Zhang [2022] argue that, in the context of distributionally robust optimization, solving Objective 1 may *not* be advantageous. Additionally, they introduced a different objective (see Objective 7) which can be favorable to minimize. In this section, we extend our guarantees for transformation-invariant learning to this new objective which we describe next.

For each $T \in \mathcal{T}$, let $\text{OPT}_T = \inf_{h_T^\star \in \mathcal{H}} \text{err}(h_T^\star, T(\mathcal{D}))$ be the smallest achievable error on transformed distribution $T(\mathcal{D})$ with hypothesis class $\mathcal{H}$. Given a training sample $S = \{(x_1, y_1), \ldots, (x_m, y_m)\} \sim \mathcal{D}^m$, we would like to learn a predictor $\hat{h} : \mathcal{X} \to \mathcal{Y}$ with *uniformly small regret* across all transformations $T$ in $\mathcal{T}$,

$$\sup_{T \in \mathcal{T}} \text{err}(\hat{h}, T(\mathcal{D})) - \text{OPT}_T \le \inf_{h^\star \in \mathcal{H}} \sup_{T \in \mathcal{T}} \left\{\text{err}(h^\star, T(\mathcal{D})) - \text{OPT}_T\right\} + \varepsilon. \tag{7}$$

We illustrate with a concrete example below how solving Objective 7 can be favorable to Objective 1.
*Example* 2 (Risk vs. Regret). Consider a class $\mathcal{H} = \{h_1, h_2\}$, a collection of transformations $\mathcal{T} = \{T_1, T_2, T_3, T_4\}$, and a distribution $\mathcal{D}$ such that with errors as reported in the table:

|       | $T_1(\mathcal{D})$ | $T_2(\mathcal{D})$ | $T_3(\mathcal{D})$ | $T_4(\mathcal{D})$ |
|-------|--------------------|--------------------|--------------------|--------------------|
| $h_1$ | 0                  | $1/8$              | $1/4$              | $1/2$              |
| $h_2$ | $1/2$              | $1/2$              | $1/2$              | $1/2$              |

Thus, solving Objective 1 may return predictor $h_2$ where $\forall T \in \mathcal{T} : \text{err}(h_2, T(\mathcal{D})) = 1/2$, since we only need to compete with the worst-case risk $\text{OPT}_\infty = \frac{1}{2}$. However, solving Objective 7 will return predictor $h_1$ where $\forall T : \mathcal{T} : \text{err}(h_1, T(\mathcal{D})) \le \text{OPT}_T$.

More generally, as highlighted by Agarwal and Zhang [2022], whenever there exists $h^\star \in \mathcal{H}$ satisfying $\forall T \in \mathcal{T} : \text{err}(h^\star, T(\mathcal{D})) - \text{OPT}_T \le \varepsilon$, solving Objective 7 is favorable. We present next a generic learning rule solving Objective 7 for any hypothesis class $\mathcal{H}$ and any collection of transformations $\mathcal{T}$,

$$\hat{h} \in \underset{h \in \mathcal{H}}{\arg\min} \max_{T \in \mathcal{T}} \left\{\frac{1}{m} \sum_{i=1}^m \mathbb{1}\left[h(T(x_i)) \ne y_i\right] - \hat{\text{OPT}}_T\right\}. \tag{8}$$

We present next a PAC-style learning guarantee for this learning rule with sample complexity bounded by the VC dimension of the composition of $\mathcal{H}$ with $\mathcal{T}$. The proof is deferred to Appendix E.

**Theorem 5.1.** *For any class $\mathcal{H}$, any collection of transformations $\mathcal{T}$, any $\varepsilon, \delta \in (0, 1/2)$, any distribution $\mathcal{D}$, with probability at least $1 - \delta$ over $S \sim \mathcal{D}^{m(\varepsilon,\delta)}$ where $m(\varepsilon, \delta) = O\left(\frac{\text{vc}(\mathcal{H} \circ \mathcal{T}) + \log(1/\delta)}{\varepsilon^2}\right)$,*

$$\sup_{T \in \mathcal{T}} \left\{\text{err}(\hat{h}, T(D)) - \text{OPT}_T\right\} \le \inf_{h^\star \in \mathcal{H}} \sup_{T \in \mathcal{T}} \left\{\text{err}(h^\star, T(\mathcal{D})) - \text{OPT}_T\right\} + \varepsilon.$$

**Algorithmic Reduction to ERM.** Using ideas and techniques similar to those used in Section 3, we develop a generic oracle-efficient reduction solving Objective 7 using only an ERM oracle for $\mathcal{H}$. Theorem, proof, and algorithm are deferred to Appendix E.

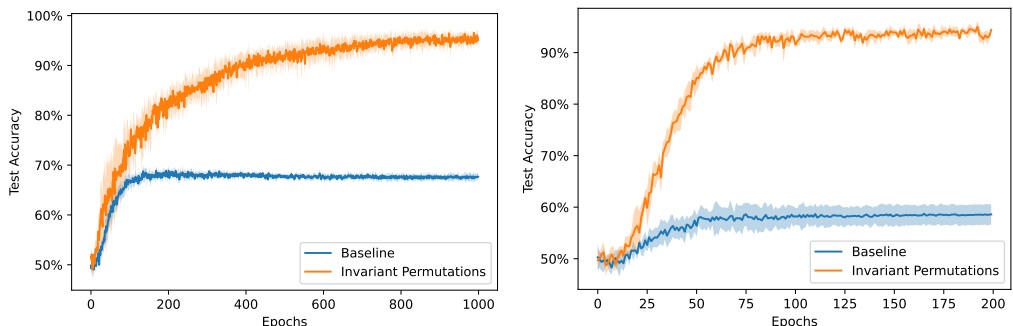

Figure 1: Left plot is for learning $f_1^\star$, the full parity function in dimension 18, with a train set size of 7000. Transformations are sampled from $\mathcal{T}_1$: the set of *all* permutations. Right plot is for learning $f_2^\star$, a majority-of-subparities function in dimension 21, with a train set size of 5000. Transformations are sampled from $\mathcal{T}_2$: permutations on which $f_2^\star$ is invariant. In each case, the test set size is 1000.

## 6 Basic Experiment

We present results for a basic experiment on learning Boolean functions on the hypercube $\{\pm 1\}^d$. We consider a uniform distribution $D$ over $\{\pm 1\}^d$ and two target functions: (1) $f_1^\star(x) = \Pi_{i=1}^d x_i$, the parity function, and (2) $f_2^\star(x) = \mathrm{sign}(\sum_{j=0}^2 (\Pi_{i=1}^{d/3} x_{j(d/3)+i}))$, a majority-of-subparities function. We consider transformations $\mathcal{T}_1, \mathcal{T}_2$ under which $f_1^\star, f_2^\star$ are invariant, respectively (see Section 2). Since $D$ is uniform, note that for any $\hat{h}$: $\sup_{T \in \mathcal{T}} \mathrm{err}(\hat{h}, T(D_{f^\star})) = \mathrm{err}(\hat{h}, D_{f^\star})$.

**Algorithms.** We use a two-layer feed-forward neural network architecture with 512 hidden units as our hypothesis class $\mathcal{H}$. We use the squared loss and consider two training algorithms. First, the baseline is running standard mini-batch SGD on training examples. Second, as a heuristic to implement Equation (2), we run mini-batch SGD on training examples and permutations of them. Specifically, in each step we replace correctly classified training examples in a mini-batch with random permutations of them (drawn from $\mathcal{T}$), and then perform an SGD update on this modified mini-batch. We set the mini-batch size to 1 and the learning rate to 0.01. Results are averaged over 5 runs with different seeds and are reported in Figure 1. We ran experiments on freely available Google CoLab T4 GPUs, and used Python and PyTorch to implement code.

## 7 Related Work and Discussion

**Covariate Shift, Domain Adaptation, Transfer Learning.** There is substantial literature studying theoretical guarantees for learning when there is a "source" distribution $P$ and a "target" distribution $Q$ [see e.g., survey by Redko et al., 2020, Quinonero-Candela et al., 2008]. Many of these works explore structural relationships between $P$ and $Q$ using various divergence measures (e.g., total variation distance or KL divergence), sometimes incorporating the structure of the hypothesis class $\mathcal{H}$ [e.g., Ben-David et al., 2010, Hanneke and Kpotufe, 2019]. Sometimes access to unlabeled (or few labeled) samples from $Q$ is assumed. Our work differs from this line of work by expressing the structural relationship between $P$ and $Q$ in terms of a transformation $T$ where $Q = T(P)$.

**Distributionally Robust Optimization.** With roots in optimization literature [see e.g., Ben-Tal et al., 2009, Shapiro, 2017], this framework has been further studied recently in the machine learning literature [see e.g., Duchi and Namkoong, 2021]. The goal is to learn a predictor $\hat{h}$ that minimizes the worst-case error $\sup_{Q \in \mathcal{P}} \mathrm{err}(\hat{h}, Q)$, where $\mathcal{P}$ is a collection of distributions. Most prior work adopting this framework has focused on distributions $\mathcal{P}$ that are close to a "source" distribution $\mathcal{D}$ in some divergence measure [e.g., $f$-divergences Namkoong and Duchi, 2016]. Instead of relying on divergence measures, our work describes the collection $\mathcal{P}$ through data transformations $\mathcal{T}$ of $\mathcal{D}$: $\{T(\mathcal{D})\}_{T \in \mathcal{T}}$ which may be operationally simpler.

**Multi-Distribution Learning.** This line of work focuses on the setting where there are $k$ arbitrary distributions $\mathcal{D}_1, \ldots, \mathcal{D}_k$ to be learned uniformly well, where sample access to each distribution $\mathcal{D}_i$ is provided [see e.g., Haghtalab et al., 2022]. In contrast, our setting involves access to a

single distribution $\mathcal{D}$ and transformations $T_1, \ldots, T_k$, that together describe the target distributions: $T_1(\mathcal{D}), \ldots, T_k(\mathcal{D})$. From a sample complexity standpoint, multi-distribution learning requires sample complexity scaling linearly in $k$ while in our case it is possible to learn with sample complexity scaling logarithmically in $k$ (see Theorem 3.1 and Lemma B.1). The lower sample complexity in our approach is primarily due to the assumption that the transformations $T_1, \ldots, T_k$ are known in advance, allowing the learner to generate $k$ samples from a single draw of $\mathcal{D}$. In contrast, in multi-distribution learning, the learner pays for $k$ samples in order to see one sample from each of $\mathcal{D}_1, \ldots, \mathcal{D}_k$. Therefore, while the sample complexity is lower in our setting, this advantage arises from the additional information/structure provided rather than an inherent improvement over the more general setting of multi-distribution learning. From an algorithmic standpoint, our reduction algorithms employ similar techniques based on regret minimization and solving zero-sum games [Freund and Schapire, 1997].

**Invariant Risk Minimization (IRM).** This is another formulation addressing domain generalization or learning a predictor that performs well across different environments [Arjovsky et al., 2019]. One main difference from our work is that in the IRM framework training examples from different environments are observed and no explicit description of the transformations is provided. Furthermore, to argue about generalization on environments unseen during training, a structural causal model is considered. Recent works have highlighted some drawbacks of IRM [Rosenfeld et al., 2021, Kamath et al., 2021]. For example, how in some cases ERM outperforms IRM on out-of-distribution generalization, and the sensitivity of IRM to finite empirical samples vs. infinite population samples.

**Data Augmentation.** A commonly used technique in learning under invariant transformations is data augmentation, which involves adding transformed data into the training set and training a model with the augmented data. Theoretical guarantees of data augmentation have received significant attention recently [see e.g., Dao et al., 2019, Chen et al., 2020, Lyle et al., 2020, Shao et al., 2022, Shen et al., 2022]. In this line of research, it is common to assume that the transformations form a group, and the learning goal is to achieve good performance under the "source" distribution by leveraging knowledge of the invariant transformations structure. In contrast, our work does not make the group assumption over transformations, and our goal is to learn a model with low loss under all possible "target" distributions parameterized by transformations of the "source" distribution.

**Multi-Task Learning.** Ben-David and Borbely [2008] studied conditions underwhich a set of transformations $\mathcal{T}$ can help with multi-task learning, assuming that $\mathcal{T}$ forms a group and that $\mathcal{H}$ is closed under $\mathcal{T}$. Our work does not make such assumptions, and studies a different learning objective.

## Acknowledgments

We thank Suriya Gunasekar for insightful discussions at early stages of this work. This work was done in part under the NSF-Simons Collaboration on the Theoretical Foundations of Deep Learning. OM was supported by a FODSI-Simons postdoctoral fellowship at UC Berkeley. HS was supported in part by Harvard CMSA. This work was conducted primarily while HS was at TTIC and supported in part by the National Science Foundation under grants CCF-2212968 and ECCS-2216899, and by the Defense Advanced Research Projects Agency under cooperative agreement HR00112020003. The views expressed in this work do not necessarily reflect the position or the policy of the Government and no official endorsement should be inferred.

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

# A   Uniform Convergence

We can use tools from VC theory [Vapnik and Chervonenkis, 1971, 1974] to obtain uniform convergence guarantees that will allow us to establish our learning guarantees and sample complexity bounds in the remainder of the paper. The starting point is a simple but key observation concerning the hypothesis class $\mathcal{H}$ and the collection of transformations $\mathcal{T}$. Specifically, consider the composition of $\mathcal{H}$ with $\mathcal{T}$ defined as:

$$\mathcal{H} \circ \mathcal{T} := \{h \circ T : h \in \mathcal{H}, T \in \mathcal{T}\}, \text{ where } (h \circ T)(x) = h(T(x)) \ \forall x \in \mathcal{X}. \tag{9}$$

We next apply VC theory to the class $\mathcal{H} \circ \mathcal{T}$ to obtain our uniform convergence guarantees. Formally,

**Proposition A.1.** *For any class $\mathcal{H}$, any collection of transformations $\mathcal{T}$, any distribution $\mathcal{D}$ over $\mathcal{X} \times \mathcal{Y}$, and any $m \in \mathbb{N}$, with probability at least $1 - \delta$ over $S \sim \mathcal{D}^m$: $\forall h \in \mathcal{H}, \forall T \in \mathcal{T}$,*

$$|\mathrm{err}(h, T(S)) - \mathrm{err}(h, T(\mathcal{D}))| \leq c\sqrt{\frac{\mathrm{vc}(\mathcal{H} \circ \mathcal{T}) + \log(1/\delta)}{m}}.$$

*Proof.* Since the composition $\mathcal{H} \circ \mathcal{T}$ is a hypothesis class consisting of functions $h \circ T$ where $h \in \mathcal{H}, T \in \mathcal{T}$, the claim follows from the definition of VC dimension and uniform convergence guarantees for VC classes [Vapnik and Chervonenkis, 1971, 1974]. □

**Proposition A.2** (Optimistic Rate). *For any class $\mathcal{H}$, any collection of transformations $\mathcal{T}$, any distribution $\mathcal{D}$, and any $m \in \mathbb{N}$, letting $B(m, \delta) := \frac{1}{m}\left(\mathrm{vc}(\mathcal{H} \circ \mathcal{T}) \log\left(\frac{2em}{\mathrm{vc}(\mathcal{H} \circ \mathcal{T})}\right) + \log(4/\delta)\right)$, with probability at least $1 - \delta$ over $S \sim \mathcal{D}^m$: $\forall h \in \mathcal{H}, \forall T \in \mathcal{T}$,*

$$\mathrm{err}(h, T(\mathcal{D})) \leq \mathrm{err}(h, T(S)) + 2\sqrt{\mathrm{err}(h, T(S))B(m, \delta)} + 4B(m, \delta),$$
$$\mathrm{err}(h, T(S)) \leq \mathrm{err}(h, T(\mathcal{D})) + 2\sqrt{\mathrm{err}(h, T(\mathcal{D}))B(m, \delta)} + 4B(m, \delta).$$

*Proof.* The claim follows from applying relative deviation bounds, or optimistic rates, for the composition class $\mathcal{H} \circ \mathcal{T}$ [see e.g., Corollary 7 in Cortes et al., 2019]. □

# B   Bounding the VC dimension of $\mathcal{H}$ composed with $\mathcal{T}$

Given the relevance of $\mathrm{vc}(\mathcal{H} \circ \mathcal{T})$ in our theoretical study, in this section we explore the relationship between $\mathrm{vc}(\mathcal{H} \circ \mathcal{T})$ and $\mathrm{vc}(\mathcal{H})$ which we believe can be helpful in interpreting our results and comparing them with the sample complexity of standard PAC learning [which is controlled by $\mathrm{vc}(\mathcal{H})$ Blumer et al., 1989a, Ehrenfeucht et al., 1989]. To this end, we consider below a few general cases and prove bounds on $\mathrm{vc}(\mathcal{H} \circ \mathcal{T})$ in terms of $\mathrm{vc}(\mathcal{H})$ and some form of capacity control for $\mathcal{T}$. These results may be of independent interest.

**Finitely many transformations.** When $\mathcal{T}$ is a *finite* collection of transformations, we can bound $\mathrm{vc}(\mathcal{H} \circ \mathcal{T})$ from above by $O(\mathrm{vc}(\mathcal{H}) + \log|\mathcal{T}|)$ using the Sauer-Shelah-Perels Lemma [Sauer, 1972],

**Lemma B.1.** *For any class $\mathcal{H}$ and any* finite *collection $\mathcal{T}$, $\mathrm{vc}(\mathcal{H} \circ \mathcal{T}) \leq O(\mathrm{vc}(\mathcal{H}) + \log|\mathcal{T}|)$.*

*Proof.* Consider an arbitrary set of points $P = \{x_1, \ldots, x_m\} \subseteq \mathcal{X}$. To bound $\mathrm{vc}(\mathcal{H} \circ \mathcal{T})$ from above, it suffices to bound the number of behaviors when projecting the function class $\mathcal{H} \circ \mathcal{T}$ on $P$, defined as

$$\Pi_{\mathcal{H} \circ \mathcal{T}}(P) := \{(h(T(x_1)), \ldots, h(T(x_m))) : h \in \mathcal{H}, T \in \mathcal{T}\}.$$

Observe that

$$|\Pi_{\mathcal{H} \circ \mathcal{T}}(P)| \leq \sum_{T \in \mathcal{T}} |\{(h(T(x_1)), \ldots, h(T(x_m))) : h \in \mathcal{H}\}| \leq |\mathcal{T}| \left(\frac{em}{\mathrm{vc}(\mathcal{H})}\right)^{\mathrm{vc}(\mathcal{H})},$$

where the first inequality follows from the definition of $\Pi_{\mathcal{H} \circ \mathcal{T}}(P)$ and the second inequality follows from the Sauer-Shelah-Perels Lemma [Sauer, 1972]. Solving for $m$ such that the above bound is less than $2^m$, implies that $\mathrm{vc}(\mathcal{H} \circ \mathcal{T}) \leq O(\mathrm{vc}(\mathcal{H}) + \log|\mathcal{T}|)$. □

**Linear transformations.** Consider $\mathcal{T}$ being a (potentially infinite) collection of *linear* transformations. For example, in vision tasks, this includes transforming images through rotations, translations, maskings, adding random noise (or any fixed a-priori arbitrary noise), and their compositions. Surprisingly, for a broad range of hypothesis classes $\mathcal{H}$ (including linear predictors and neural networks), we can show that $\mathrm{vc}(\mathcal{H} \circ \mathcal{T}) \leq \mathrm{vc}(\mathcal{H})$ without incurring any dependence on the complexity of $\mathcal{T}$. Specifically, the result applies to any function class $\mathcal{H}$ that consists of a linear mapping followed by an arbitrary mapping. This includes feed-forward neural networks with any activation function, and modern neural network architectures (e.g., CNNs, ResNets, Transformers).

**Lemma B.2.** *For any collection of* linear *transformations $\mathcal{T}$ and any hypothesis class of the form $\mathcal{H} = \{f \circ W : \mathbb{R}^d \to \mathcal{Y} \mid W \in \mathbb{R}^{k \times d} \wedge f : \mathbb{R}^k \to \mathcal{Y}\}$, it holds that $\mathrm{vc}(\mathcal{H} \circ \mathcal{T}) \leq \mathrm{vc}(\mathcal{H})$.*

*Proof.* By definition of the VC dimension, it suffices to show that $\mathcal{H} \circ \mathcal{T} \subseteq \mathcal{H}$. To this end, consider an arbitrary $h = f \circ W \in \mathcal{H}$ where $W : \mathbb{R}^d \to \mathbb{R}^k$ is a linear map and $f : \mathbb{R}^k \to \mathcal{Y}$ is an arbitrary map (see definition of $\mathcal{H}$ in the lemma statement), and consider an arbitrary linear transformation $T \in \mathcal{T}$. Then, observe that for each $x \in \mathbb{R}^d$,

$$(h \circ T)(x) = (f \circ W)(T(x)) = f(W(T(x))) = f(T^*(W)(x)),$$

where the last equality follows from Riesz Representation theorem and $T^*$ is the adjoint transformation of $T$. Thus, we have shown that there exists $\tilde{W} = T^*(W)$ such that $(f \circ W)(T(x)) = (f \circ \tilde{W})(x)$ for all $x \in \mathbb{R}^d$. Therefore, $\mathcal{H} \circ \mathcal{T} \subseteq \mathcal{H}$. $\qquad\square$

**Transformations on the Boolean hypercube.** When the instance space $\mathcal{X} = \{0, 1\}^d$ is the Boolean hypercube, we can bound the VC dimension of $\mathcal{H} \circ \mathcal{T}$ from above by the sum of the VC dimension of $\mathcal{H}$ and the sum of the VC dimensions of $\{\mathcal{T}_i\}_{i=1}^d$ where each $\mathcal{T}_i = \{x \mapsto T(x)_i : T \in \mathcal{T}\}$ is a function class resulting from restricting transformations $T : \{0,1\}^d \to \{0,1\}^d \in \mathcal{T}$ to output only the $i$th bit.

**Lemma B.3.** *When $\mathcal{X} = \{0,1\}^d$, for any hypothesis class $\mathcal{H}$ and any collection of transformations $\mathcal{T}$, $\mathrm{vc}(\mathcal{H} \circ \mathcal{T}) \leq O(\log d)(\mathrm{vc}(\mathcal{H}) + \sum_{i=1}^d \mathrm{vc}(\mathcal{T}_i))$, where each $\mathcal{T}_i = \{x \mapsto T(x)_i : T \in \mathcal{T}\}$.*

*Proof.* Every function $h \circ T \in \mathcal{H} \circ \mathcal{T}$ can be viewed as $x \mapsto h(T(x)_1, \ldots, T(x)_d)$, which is a composition of $h$ with $d$ Boolean functions $T(\cdot)_1, \ldots, T(\cdot)_d : \mathcal{X} \to \{0,1\}$ where each $T(\cdot)_i$ is the restriction of transformation $T$ to the $i$th coordinate. The claim then follows from a direct application of Proposition 4.9 in Alon et al. [2023], which itself generalizes a classical result due to Blumer et al. [1989b] bounding the VC dimension of composed function classes. $\qquad\square$

## C   Proofs for Section 2

*Proof of Theorem 2.2.* We note that the proof follows a standard no-free-lunch argument for VC classes, where the game will be to guess the support of a distribution.

Let $x_1, \ldots, x_{3k}$ be arbitrary points. For each $P \subseteq [3k]$ where $|P| = k$, define a transformation $T_P$ that maps $x_1, \ldots, x_{3k}$ to some new and unique points $T_P(x_1), \ldots, T_P(x_{3k})$. Then, define classifier $h_P$ to be positive everywhere, except on the points $\{T_P(x_i)\}_{i \in P}$ which are labeled negative. Let $\mathcal{X} = \{x_1, \ldots, x_{3k}\} \bigcup_P \{T_P(x_1), \ldots, T_P(x_{3k})\}$, $\mathcal{H} = \{h_P : P \subseteq [3k], |P| = k\}$, and $\mathcal{T} = \{T_P : P \subseteq [3k], |P| = k\}$.

It is easy to see that $\mathrm{vc}(\mathcal{H}) = 1$, since classifiers in $\mathcal{H}$ operate in different parts of $\mathcal{X}$. Furthermore, $\mathrm{vc}(\mathcal{H} \circ \mathcal{T}) \geq k$ where we can shatter $x_1, \ldots, x_k$ with $\mathcal{H} \circ \mathcal{T}$ as follows: for each $y_1, \ldots, y_k$, let $I = \{i \in [k] : y_i = -1\}$ and $P = I \cup \{j : k + 1 \leq j \leq 2k - |I|\}$, then $(h_P \circ T_P)(x_i) = h_P(T_P(x_i)) = y_i$ for all $i \in [k]$.

Consider now a family of distributions $\{\mathcal{D}_P : P \subseteq [3k], |P| = k\}$ over $\mathcal{X} \times \mathcal{Y}$ where each $\mathcal{D}_P$ is uniform over $2k$ points $\{(x_i, +1)\}_{i \notin P}$. For each $P \subseteq [3k]$ where $|P| = k$, observe that by definitions of $\mathcal{D}_P, \mathcal{H}, \mathcal{T}$, $\sup_{T \in \mathcal{T}} \mathrm{err}(h_P, T(\mathcal{D}_P)) = 0$ since $h_P$ only labels the points $\{T_P(x_i)\}_{i \in P}$ negative and $\{x_i\}_{i \in P}$ are not in the support of $\mathcal{D}_P$. That is to say, our lower bound holds in the realizable setting where $\mathsf{OPT}_\infty = 0$ (see Equation (1)).

Next, consider an arbitrary proper learning rule $\mathbb{A} : (\mathcal{X} \times \mathcal{Y})^* \to \mathcal{H}$. For a distribution $\mathcal{D}_P$ chosen uniformly at random and upon receiving a random sample $S \sim \mathcal{D}_P^k$, $\mathbb{A}$ needs to *correctly* guess which points from $\{x_1, \ldots, x_{3k}\} \setminus S$ lie in the support of $\mathcal{D}_P$ in order to choose an appropriate $h \in \mathcal{H}$ with small error. However, since the support is chosen uniformly at random, $\mathbb{A}$ will most likely incorrectly guess a constant fraction of the support, leading to a constant error. This is a standard argument [see e.g., Montasser et al., 2019, Alon et al., 2021], but we repeat it below for completeness.

Fix an arbitrary sequence $S \in \{(x_1, +1), \ldots, (x_{3k}, +1)\}^k$. Denote by $E_S$ the event that $S \in \text{supp}(\mathcal{D}_P)$ for a distribution $\mathcal{D}_P$ that is picked uniformly at random. Next,

$$
\begin{aligned}
\mathbb{E}_P \left[ \sup_{T \in \mathcal{T}} \text{err}(\mathbb{A}(S), T(\mathcal{D}_P)) | E_S \right] &\geq \mathbb{E}_P \left[ \text{err}(\mathbb{A}(S), T_P(\mathcal{D}_P)) | E_S \right] \\
&\geq \mathbb{E}_P \left[ \frac{1}{2k} \sum_{i \notin P} \mathbb{1}[\mathbb{A}(S)(T_P(x_i)) \neq +1] | E_S \right] \\
&\geq \frac{1}{2} \mathbb{E}_P \left[ \frac{1}{k} \sum_{i \notin P \wedge (x_i, +1) \notin S} \mathbb{1}[\mathbb{A}(S)(T_P(x_i)) \neq +1] | E_S \right] \\
&\geq \frac{1}{4},
\end{aligned}
$$

where the last inequality follows from the fact that $\mathbb{A}(S) \in \mathcal{H}$ and that the remaining (at least $k$) points that are not in $S$ but in $\text{supp}(\mathcal{D}_P)$ are chosen uniformly at random, because $\mathcal{D}_P$ is chosen randomly. From the above, by law of total expectation, we have

$$
\mathbb{E}_P \mathbb{E}_{S \sim \mathcal{D}_P^k} \left[ \sup_{T \in \mathcal{T}} \text{err}(\mathbb{A}(S), T(\mathcal{D}_P)) \right] \geq \frac{1}{4}.
$$

By the probabilistic method, this means there exists $P^*$ such that $\mathbb{E}_{S \sim \mathcal{D}_{P^*}^k} \left[ \sup_{T \in \mathcal{T}} \text{err}(\mathbb{A}(S), T(\mathcal{D}_P)) \right] \geq \frac{1}{4}$. Using a variant of Markov's inequality, this implies that $\Pr_{S \sim \mathcal{D}_{P^*}^k} \left[ \sup_{T \in \mathcal{T}} \text{err}(\mathbb{A}(S), T(\mathcal{D}_P)) > \frac{1}{8} \right] \geq \frac{1}{7}$. $\quad\square$

## D  Proofs for Section 3

**Proposition D.1.** *For any class $\mathcal{H}$, any* ERM *for $\mathcal{H}$, any collection of transformations $\mathcal{T}$, any distribution $\mathcal{D}$ such that $\text{OPT}_\infty = 0$, and any $\varepsilon, \delta \in (0, 1/2)$, with probability at least $1 - \delta$ over $S \sim \mathcal{D}^{m(\varepsilon, \delta)}$, where $m(\varepsilon, \delta) = O\left( \frac{\text{vc}(\mathcal{H} \circ \mathcal{T}) \log(1/\varepsilon) + \log(1/\delta)}{\varepsilon} \right)$,*

$$
\forall T \in \mathcal{T} : \text{err}(\hat{h}, T(\mathcal{D})) \leq \varepsilon,
$$

*where $\hat{h}$ is the output predictor of running* ERM *on the inflated dataset $\mathcal{T}(S) = \{(T(x), y) : (x, y) \in S \wedge T \in \mathcal{T}\}$.*

*Proof of Proposition D.1.* Since $\text{OPT}_\infty = 0$, i.e., there exists $h^\star \in \mathcal{H}$ such that $\forall T \in \mathcal{T} : \text{err}(h^\star, T(\mathcal{D})) = 0$, and since $\hat{h}$ is the output predictor of running ERM on the inflated dataset $\mathcal{T}(S) = \{(T(x), y) : (x, y) \in S \wedge T \in \mathcal{T}\}$, it follows that $\forall T \in \mathcal{T} : \text{err}(\hat{h}, T(S)) = 0$. Thus, by invoking the optimistic generalization guarantee (Proposition A.2), with probability at least $1 - \delta$ over $S \sim \mathcal{D}^{m(\varepsilon, \delta)}$: $(\forall h \in \mathcal{H})(\forall T \in \mathcal{T}) : \text{err}(h, T(S)) \Rightarrow \text{err}(h, T(\mathcal{D})) \leq \varepsilon$. Since $\hat{h} \in \mathcal{H}$, it follows that $\forall T \in \mathcal{T} : \text{err}(\hat{h}, T(\mathcal{D})) \leq \varepsilon$. $\quad\square$

**Lemma D.2.** *Let $S = \{(x_1, y_1), \ldots, (x_m, y_m)\}$ be an arbitrary dataset. For any distribution $Q$ over $\mathcal{T}$ and any predictor $h$, define the loss function $\ell_S(h, Q) = 1 - \mathbb{E}_{T \sim Q} \text{err}(h, T(S))$. Then for any sequence of predictors $h_1, \ldots, h_R$, running Multiplicative Weights with $\eta = \sqrt{8 \ln |\mathcal{T}| / R}$ (see Algorithm 1) produces a sequence of distributions $Q_1, \ldots, Q_R$ over $\mathcal{T}$ that satisfy*

$$
\frac{1}{R} \sum_{r=1}^R \ell_S(h_r, Q_r) \leq \min_{T \in \mathcal{T}} \frac{1}{R} \sum_{r=1}^R \ell_S(h_r, T) + \sqrt{\frac{\ln |\mathcal{T}|}{2R}}.
$$

*Proof of Lemma D.2.* The proof follows directly from considering a two-player game, where the $\mathcal{T}$-player plays mixed strategies (distributions over $\mathcal{T}$) $Q_1, \ldots, Q_R$ against predictors $h_1, \ldots, h_R$ played by an arbitrary learning algorithm $\mathbb{A}$, and in each round the $\mathcal{T}$-player incurs loss $\ell_S(h_r, Q_r) = 1 - \mathbb{E}_{T \sim Q_r} \mathrm{err}(h_r, T(S))$. Then, the regret guarantee of Multiplicative Weights [see e.g., Theorem 2.2 in Cesa-Bianchi and Lugosi, 2006] implies that

$$\frac{1}{R} \sum_{r=1}^{R} \ell_S(h_r, Q_r) \leq \min_{T \in \mathcal{T}} \frac{1}{R} \sum_{r=1}^{R} \ell_S(h_r, T) + \sqrt{\frac{\ln|\mathcal{T}|}{2R}}. \qquad \square$$

# E Proofs for Section 5

*Proof of Theorem 5.1.* The proof follows from the definition of $\hat{h}$ (Equation (8)) and using uniform convergence bounds (Proposition A.1). Let $h^\star \in \mathcal{H}$ be an a-priori fixed predictor (independent of sample $S$) attaining

$$\inf_{h^\star \in \mathcal{H}} \sup_{T \in \mathcal{T}} \{\mathrm{err}(h^\star, T(\mathcal{D})) - \mathsf{OPT}_T\}$$

or is $\varepsilon$-close to it. By setting $m(\varepsilon, \delta) = m(\varepsilon, \delta) = O\left(\frac{\mathrm{vc}(\mathcal{H} \circ \mathcal{T}) + \log(1/\delta)}{\varepsilon^2}\right)$ and invoking Proposition A.1, we have the guarantee that with probability at least $1 - \delta$ over $S \sim \mathcal{D}^{m(\varepsilon, \delta)}$,

$$(\forall h \in \mathcal{H}) \, (\forall T \in \mathcal{T}) : |\mathrm{err}(h, T(S)) - \mathrm{err}(h, T(\mathcal{D}))| \leq \varepsilon.$$

Since $\hat{h}, h^\star \in \mathcal{H}$, the inequality above implies that

$$\forall T \in \mathcal{T} : \; \mathrm{err}(\hat{h}, T(\mathcal{D})) \leq \mathrm{err}(\hat{h}, T(S)) + \varepsilon.$$
$$\forall T \in \mathcal{T} : \; \mathrm{err}(h^\star, T(S)) \leq \mathrm{err}(h^\star, T(\mathcal{D})) + \varepsilon.$$
$$\forall T \in \mathcal{T} : \; \left|\mathsf{OPT}_T - \hat{\mathsf{OPT}}_T\right| \leq \varepsilon.$$

Furthermore, by definition, since $\hat{h}$ minimizes the empirical objective, it holds that

$$\sup_{T \in \mathcal{T}} \mathrm{err}(\hat{h}, T(S)) - \hat{\mathsf{OPT}}_T \leq \sup_{T \in \mathcal{T}} \mathrm{err}(h^\star, T(S)) - \hat{\mathsf{OPT}}_T.$$

By combining the above, we get

$$\forall T \in \mathcal{T} : \mathrm{err}(\hat{h}, T(\mathcal{D})) - \mathsf{OPT}_T \leq \sup_{T \in \mathcal{T}} \mathrm{err}(\hat{h}, T(S)) - \hat{\mathsf{OPT}}_T + 2\varepsilon$$
$$\leq \sup_{T \in \mathcal{T}} \mathrm{err}(h^\star, T(S)) - \hat{\mathsf{OPT}}_T + 2\varepsilon$$
$$\leq \sup_{T \in \mathcal{T}} \mathrm{err}(h^\star, T(S)) - \mathsf{OPT}_T + 3\varepsilon.$$

This concludes the proof by definition of $h^\star$. $\qquad \square$

Similar to Section 3, we develop in this section a generic oracle-efficient reduction solving Objective 7 using only an ERM oracle for $\mathcal{H}$. This reduction may be favorable in applications where we only have black-box access to an off-the-shelve supervised learning method. The techniques used are similar to those used in Section 3, and Agarwal and Zhang [2022] who developed a similar reduction when having access to a collection of importance weights instead of a collection of transformations (which is the view we propose in this work).

**Theorem E.1.** *For any class $\mathcal{H}$, collection of transformations $\mathcal{T}$, distribution $\mathcal{D}$ and any $\varepsilon, \delta \in (0, 1/2)$, with probability at least $1 - \delta$ over $S \sim \mathcal{D}^{m(\varepsilon, \delta)}$, where $m(\varepsilon, \delta) \leq O\left(\frac{\mathrm{vc}(\mathcal{H} \circ \mathcal{T}) + \log(1/\delta)}{\varepsilon^2}\right)$, running Algorithm 2 on $S$ for $R \geq \frac{8 \ln|\mathcal{T}|}{\varepsilon^2}$ rounds produces $\bar{h} = \frac{1}{R} \sum_{r=1}^{R} h_r$ s.t.*

$$\forall T \in \mathcal{T} : \; \mathrm{err}(\bar{h}, T(D)) - \mathsf{OPT}_T \leq \inf_{h^\star \in \mathcal{H}} \sup_{T \in \mathcal{T}} \{\mathrm{err}(h^\star, T(\mathcal{D})) - \mathsf{OPT}_T\} + \varepsilon.$$

**Algorithm 2:** Reduction to Minimize Worst-Case Regret

**Input:** Black-box learner $\text{ERM}_{\mathcal{H}}$, dataset $S = \{(x_1, y_1), \ldots, (x_m, y_m)\}$, and transformations $\mathcal{T}$.

1   For each $T \in \mathcal{T}$, run learner $\text{ERM}_{\mathcal{H}}$ on $T(S)$ and denote its output by $\hat{h}_T$.

2   For each $T \in \mathcal{T}$, set $Q_1(T) = \frac{1}{|\mathcal{T}|}$.

3   Set $R = \frac{8 \ln |\mathcal{T}|}{\varepsilon^2}$.

4   **for** $1 \leq r \leq R$ **do**

5      Draw $m_{\text{ERM}}$ i.i.d. samples $(X_1, Y_1), \ldots, (X_{m_{\text{ERM}}}, Y_{m_{\text{ERM}}})$, where each $(X_i, Y_i)$ is drawn by randomly drawing a transformation $T$ according to $Q_t$ and randomly drawing $(X, Y)$ from $\text{Unif}(S)$, and letting $(X_i, Y_i) = (T(X), Y)$.

6      Run learner $\text{ERM}_{\mathcal{H}}$ on $(X_1, Y_1), \ldots, (X_{m_{\text{ERM}}}, Y_{m_{\text{ERM}}})$, and let $h_r$ denote its output.

7      For each $T \in \mathcal{T}$, update $Q_{r+1}(T) = \frac{Q_r(T) \exp\left(\eta(\text{err}(h_r, T(S)) - \text{err}(\hat{h}_T, T(S)))\right)}{Z_r}$, where $Z_r$ is a normalization factor such that $Q_{r+1}$ is a distribution.

**Output:** $\frac{1}{R} \sum_{r=1}^{R} h_r$.

---

*Proof of Theorem E.1.* Let $S = \{(x_1, y_1), \ldots, (x_m, y_m)\}$ be an arbitrary dataset, and let $\mathbb{A}$ be an $(\varepsilon, \delta)$-agnostic-PAC-learner for $\mathcal{H}$. Then, by setting $R \geq \frac{8 \ln |\mathcal{T}|}{\varepsilon^2}$ and invoking Lemma D.2, we are guaranteed that Algorithm 2 produces a sequence of distributions $Q_1, \ldots, Q_T$ over $\mathcal{T}$ that satisfy

$$\sup_{T \in \mathcal{T}} \frac{1}{R} \sum_{r=1}^{R} \text{err}(h_r, T(S)) - \text{err}(\hat{h}_T, T(S)) \leq \frac{1}{R} \sum_{r=1}^{R} \mathop{\mathbb{E}}_{T \sim Q_r} \left[ \text{err}(h_r, T(S)) - \text{err}(\hat{h}_T, T(S)) \right] + \varepsilon.$$

At each round $r$, observe that Step 3 in Algorithm 1 draws an i.i.d. sample from a distribution $P_r$ over $\mathcal{X} \times \mathcal{Y}$ that is defined by $Q_r$ over $\mathcal{T}$ and $\text{Unif}(S)$, and since $\text{ERM}_{\mathcal{H}}$ is an $(\varepsilon, \delta)$-agnostic-PAC-learner for $\mathcal{H}$, Steps 5-6 guarantee that

$$\mathop{\mathbb{E}}_{T \sim Q_r} \text{err}(h_r, T(S)) = \mathop{\mathbb{E}}_{T \sim Q_r} \frac{1}{m} \sum_{i=1}^{m} \mathbb{1} \{h_r(T(x_i)) \neq y_i\} \leq \inf_{h \in \mathcal{H}} \mathop{\mathbb{E}}_{T \sim Q_r} \text{err}(h, T(S)) + \varepsilon.$$

Combining the above inequalities implies that

$$\sup_{T \in \mathcal{T}} \text{err}(\bar{h}, T(S)) - \text{err}(\hat{h}_T, T(S)) \leq \frac{1}{R} \sum_{r=1}^{R} \inf_{h \in \mathcal{H}} \mathop{\mathbb{E}}_{T \sim Q_r} \left[ \text{err}(h, T(S)) - \text{err}(\hat{h}_T, T(S)) \right] + \varepsilon + \varepsilon$$

$$\leq \inf_{h \in \mathcal{H}} \sup_{T \in \mathcal{T}} \left[ \text{err}(h, T(S)) - \text{err}(\hat{h}_T, T(S)) \right] + 2\varepsilon.$$

Finally, by appealing to uniform convergence over $\mathcal{H}$ and $\mathcal{T}$, with probability at least $1 - \delta$ over $S \sim \mathcal{D}^m$, we have

$$\forall T \in \mathcal{T}: \quad \text{err}(\hat{h}_T, T(S)) \leq \text{err}(h_T^\star, T(S)) \leq \text{OPT}_T + \varepsilon,$$

$$\text{OPT}_T \leq \text{err}(\hat{h}_T, T(\mathcal{D})) \leq \text{err}(\hat{h}_T, T(S)) + \varepsilon.$$

Thus,

$$\forall T \in \mathcal{T}: \text{err}(\bar{h}, T(\mathcal{D})) - \text{OPT}_T \leq \text{err}(\bar{h}, T(S)) - \text{err}(\hat{h}_T, T(S)) + 2\varepsilon$$

$$\leq \inf_{h \in \mathcal{H}} \sup_{T \in \mathcal{T}} \left[ \text{err}(h, T(S)) - \text{err}(\hat{h}_T, T(S)) \right] + 4\varepsilon$$

$$\leq \inf_{h \in \mathcal{H}} \sup_{T \in \mathcal{T}} [\text{err}(h, T(\mathcal{D})) - \text{OPT}_T] + 6\varepsilon. \qquad \square$$

