# OpenReview forum: "Transformation-Invariant Learning and Theoretical Guarantees for OOD Generalization"
_NeurIPS.cc/2024/Conference — NeurIPS 2024 poster_

### Official Review · Reviewer_fSmB · 2024-07-01

**Soundness:** 4
**Presentation:** 4
**Contribution:** 3
**Rating:** 7
**Confidence:** 4

**Summary:**

The authors study a new framework for learning under distribution shift caused by a family of `invariance maps’ $\mathscr{T}: X \to X$. Informally, one may think of this as generalizing e.g. settings such as learning under invariance to rotation, where the maps may be rotation matrices and the learner should perform well at test time regardless of whether the raw data has been rotated.

More formally, the learner is given a hypothesis class $H$, a family of maps $\mathscr{T}$, and sample access to a distribution $D$ over $X \times {0,1}$, and wishes to train a classifier that minimizes the test error over all possible shifts $T(D)$ for $T \in \mathscr{T}$, either globally in the sense of outputting $h$ satisfying

$$sup_T \ \{ err(h,T(D))\} \leq OPT + \varepsilon$$

Where $OPT=min_{h_* \in H} sup_T err(h^*,T(D))$, or where error is measured with respect to the best possible over each individual T (the authors call this regret minimizing):

$$sup_T \ (err(h,T(D))-OPT_T) \leq min_{h^* \in H} \ (sup_T \ (err(h^*,T(D))-OPT_T)) + \varepsilon$$

This model captures a number of settings considered in the literature. In particular, under reasonable assumptions on distributions $D,D’$ over $X$, there is always a map $T$ such that $D=T(D’)$ (Breiman’s theorem), so the setup does not lose substantial generality over classical study of covariate shift. It may also model transformation invariances as mentioned above, as well as certain types of adversarial attacks (where the adversarial shift is chosen before the adversary sees the test example).

The main results concern the sample complexity of learning in this framework in the case were 1) H is known to the learner 2) H is unknown and the learner only has ERM access to H.

In the first setting, the authors show the sample complexity of (proper) agnostic PAC learning in this model is controlled by the VC dimension of the composed class VC(H \circ T). In other words, it takes at least $VC(H \circ T)$ samples to properly learn the class, and for any $k$ there exists a VC 1 class H and maps T with $VC(H \circ T)=k$ which requires $\Omega(k)$ samples to learn.

To complement their result, the authors also investigate $VC(H \circ T)$ for a few special cases, including when $H$ is a composition $f \circ W$ of an arbitrary linear map W and any fixed $f: X \to \{0,1\}$ and $T$ is a family of linear maps (in which case $VC(H \circ T) \leq VC(H)$), some basic non-linear $T$ coming from neural net architectures, and transformations on the hypercube.

When $H$ is unknown, the authors give a similar result in the realizable case using data augmentation, and a weaker variant for the agnostic setting for finite $\mathscr{T}$ which ensures low error across a probabilistic mixture of hypotheses from H via an MWU type strategy.

Finally the authors also consider a related setting in which the learner is in some sense allowed to ignore transforms with bad error, and the learner tries to return a hypothesis with low error across as many of the transforms $T \in \mathscr{T}$ as possible.

At a technical level, most of the results are the consequence of the basic duality of transforming input data and transforming the hypothesis class, allowing the authors to move between $T(D)$ and $H \circ T$.

**Strengths:**

The authors introduce a clean theoretical framework for various types of distribution shift and transformation invariance studied in the literature, capturing several interesting problems in the literature. The proposed framework should be of interest to the theory audience at NeurIPS, and is particularly nice in that it allows easy application of classical methods in VC theory to the problem.

The authors give a (weak) characterization of its statistical complexity in terms VC(H \circ T), a natural combinatorial parameter of the concept and transformation class.

The authors discuss several natural scenarios in which their upper bound gives new sample complexity bounds on transformation invariant learning, including settings occurring in modern neural net architectures (namely the composition of linear transformations with activation functions).

The authors study several natural variants of the framework (global risk, regret, maximizing #transformations with low error) with similar corresponding upper bounds.

**Weaknesses:**

On the other hand, beyond the main upper bound and examples given by the authors, many of the further results feel somewhat incomplete. For instance:

The lower bound based on VC(H \circ T) is weaker than what one typically hopes for in a combinatorial characterization. Namely while the authors show there *exist* classes for which this bound is tight, a strong lower bound (as holds for standard learning) would show this is tight for *every* class. Is there a parameter characterizing the model in this stronger sense? The lower bound also only holds for proper learners.

In the setting where H is unknown, the use of data augmentation is a bit troubling in the sense that the size of the augmented sample could be extremely large (indeed infinite for infinite T). This isn’t captured by the model the authors introduce which simply assumes the ERM oracle can handle a call of any size, but this does not really seem to capture the computational cost of learning even at an oracle level. Furthermore, the agnostic guarantees in this setting only hold for finite $T$. POST REBUTTAL: The authors have included some discussion re: the first point that helps ameliorate the issue (e.g. using boosting one can lower the batch size to ERM at the cost of log(T) calls).

Combined with the fact that the authors main techniques are fairly standard (note in some cases, as mentioned in the strengths, it is an *advantage* of the framework one can use standard techniques! But this is less the case when these techniques do not seem to give the desired results), these issues prevent me from a stronger recommendation of the paper.

**Questions:**

See above

**Limitations:**

Yes

---

> ### Author Rebuttal · Authors · 2024-08-04
>
> We thank you for your detailed review. We address your comments below.
>
> > When $H$
>  is unknown, the authors give a similar result in the realizable case using data augmentation, and a weaker variant for the agnostic setting for finite
>  which ensures low error non-uniformly (that is, for each fixed map T, the strategy will do well, but this may not be the case for all T simultaneously as above) across a probabilistic mixture of hypotheses from H via an MWU type strategy.
>
> We want to clarify and emphasize that in Theorem 3.1, we learn a mixture strategy (a randomized predictor) with a low error guarantee that holds simultaneously across all T (Exactly the same guarantee in Theorem 2.1).
>
> > The lower bound based on VC(H \circ T) is weaker than what one typically hopes for in a combinatorial characterization. Namely while the authors show there exist classes for which this bound is tight, a strong lower bound (as holds for standard learning) would show this is tight for every class. Is there a parameter characterizing the model in this stronger sense? The lower bound also only holds for proper learners.
>
> This is a great point, and it is an interesting open question for future work to have a full characterization of what is learnable in our model in terms of $\mathcal{H}$ and $\mathcal{T}$.
>
> As mentioned in Lines 117–124, we think that improper learners could achieve better sample complexity but this remains an open question. One of the main reasons for including the lower bound (Theorem 2.2), is to argue that the sample complexity of the *simple* proper learner in Equation 2 is tight in general, and we can not hope to improve the sample complexity of this proper learner further. While there could be complex improper learners with better sample complexity, we believe that having a proper learner that has a simple description (similar to the ERM principle) is valuable even when it does not achieve the optimal sample complexity.
>
> > In the setting where H is unknown, the use of data augmentation is a bit troubling in the sense that the size of the augmented sample could be extremely large (indeed infinite for infinite T). This isn’t captured by the model the authors introduce which simply assumes the ERM oracle can handle a call of any size, but this does not really seem to capture the computational cost of learning even at an oracle level.
>
> This is a great point. We argue that for the purpose of achieving the guarantee in Equation (1) in the realizable case, it is necessary to call ERM on the *full* augmented sample. Consider an example of a distribution supported on a single point $(x,-)$ on the Real line where $x = 5$, consider transformations of $x$: $z_i = x+i$ for all $i\geq 1$ induced by some class $\mathcal{T}$. Now, calling ERM on a finite subset of these transformations will return a predictor that labels $x, z_1, …, z_{k}$ with a label $-$, and labels $z_{k+1}, \dots$ with +, which fails to satisfy our objective.
>
> Perhaps we should clarify that we are considering finite transformations (similar to the agnostic case). In which case, the computational cost of a single ERM call on $m*|T|$ samples, grows by a factor of $|T|$. But, with a fairly standard boosting argument, we can also run a classical boosting algorithm that makes multiple calls to ERM. Specifically, we run boosting starting with a uniform distribution on the *augmented sample*. In this case, each ERM call is on samples of size $O({\rm vc}(\mathcal{H}))$, and we make at most $O(\log(m*|T|))$ many ERM calls. So, we have a tradeoff between the size of a dataset given to ERM on a single call, and the total number of calls to ERM.
>
> > Furthermore, the agnostic guarantees in this setting only hold for finite $T$, and hold in the seemingly much weaker non-uniform way mentioned above. It is fairly unclear if either of these constraints are actually necessary.
>
> It is an interesting direction to extend the result in Theorem 3.1 to classes with infinite $\mathcal{T}$, perhaps with a certain parametric structure.

---

> > ### Comment · Reviewer_fSmB · 2024-08-07
> > **Rebuttal Response**
> >
> > We thank the authors' for their clarifications. The discussion of ERM complexity is enlightening, and I think is worth including in the paper (even if just in discussion).
> >
> > I think I was confused on the quantifier in Theorem 3.1, and the authors are correct; sorry! I will update my review to reflect this.

---

> > > ### Author Response · Authors · 2024-08-09
> > > **Response to reviewer fSmB07**
> > >
> > > Thank you for your feedback and for updating your review! We’re glad the discussion on ERM complexity was helpful, and we’ll include it in the paper. We appreciate your thoughtful input throughout the process.

---

### Official Review · Reviewer_RtrA · 2024-07-12

**Soundness:** 2
**Presentation:** 2
**Contribution:** 2
**Rating:** 4
**Confidence:** 3

**Summary:**

This paper studies a theoretical model of out-of-distribution (OOD) generalization, where the possible distribution shifts are encoded by a collection $\mathcal{T}$ of transformations on the domain. The goal in this setting is given a distribution $D$ to learn a hypothesis $h$ from some hypothesis class $\mathcal{H}$, such that $h$ achieves the best possible risk on the distribution $T(D)$ under the worst-case shift $T\in\mathcal{T}$. The paper gives algorithms achieving this goal in a variety of settings including when $\mathcal{H}$ and $\mathcal{T}$ are both known, and when given an ERM oracle for $\mathcal{H}$ and $\mathcal{T}$ is finite. The bounds are given in terms of the VC-dimension of the composition $\mathcal{H}\circ\mathcal{T},$ consisting of all function of the form $h\circ T$.

**Strengths:**

1. The model introduced of a class of transformations corresponding to distribution shifts gives very mathematically clean results for sample complexity in terms of the VC-dimension of the composed class $\mathcal{H}\circ\mathcal{T}$.

2. The explanations of how the model can be applied to different practical settings involving distribution shift are clear and show broad applicability.

**Weaknesses:**

The primary weakness is in distinguishing these results from prior work, particularly that on multi-distribution learning. In particular,  when the class of transformations $\mathcal{T}$ is finite, then every problem instance from this paper's model is also an instance of multi-distribution learning with $D_1,...,D_k$ given by $T_1(D),...,T_k(D)$.

Furthermore, as pointed out on line 41, if the domain of the learning problem is $\mathbb{R}^d$ then Brenier's Theorem implies that for any instance of a multi-distribution learning problem $D_1,...,D_k$ with sufficiently "nice" distributions, there is a distribution $D$ and set of $k$ transformation $T_1,\dots,T_k$ such that $D_i = T_i(D)$. Thus, in the case of finite $\mathcal{T}$ on domain $\mathbb{R}^d$ these two models are equivalent so long as only "nice" distributions are considered. The setting of finite $\mathcal{T}$ is particularly important, as it is the setting for which the main results in this paper give oracle-efficient algorithms (given an ERM oracle for $\mathcal{H}$).

The above discussion leads to what may be an issue with Theorem 3.1 as stated. As far as I can tell, Theorem 3.1 combined with Remark 3.2 would imply a $O(d + \log k)$ upper bound on sample complexity in the case that the VC-dimension of $\mathcal{H}$ is $d$ and $|\mathcal{T}|=k$. However, prior work on multi-distribution learning [1] suggests that in the setting of $k$ distributions $D_1,...,D_k$ and a hypothesis class of VC-dimension $d$ there is a lower-bound on sample complexity of $\Omega(d + k\log k)$. Perhaps I have missed something here, but I don't see any obstruction to instantiating the lower-bound family of instances from [1] as a family of instances in the model of this paper. So there seems to be some discrepancy between the known lower bounds and Theorem 3.1.

[1] Haghtalab, Nika, Michael Jordan, and Eric Zhao. "On-demand sampling: Learning optimally from multiple distributions." Advances in Neural Information Processing Systems 35 (2022): 406-419.

**Questions:**

1. Could you explain the main differences between this model and that of multi-distribution learning? In particular, are there some natural instances of multi-distribution learning which do not fall into this model?

2. As described above under weaknesses, why do the lower bounds for multi-distribution learning not apply to the results of Theorem 3.1? Have I misunderstood something about the model in this paper, or perhaps the lower bound instances from [1]?

**Limitations:**

Yes.

---

> ### Author Rebuttal · Authors · 2024-08-04
>
> We thank you for your detailed review. We clarify below how our work is different from multi-distribution  learning, and how the lower bounds on multi-distribution learning **do not** contradict our upper bound result in Theorem 3.1 / Remark 3.2.
>
> > Could you explain the main differences between this model and that of multi-distribution learning? In particular, are there some natural instances of multi-distribution learning which do not fall into this model?
>
> We summarize the main differences below:
>
> In our model, the learning algorithm has access to samples from a source distribution $D$ and access to transformations $T_1, …, T_k$. Thus, by drawing a **single** sample $(x,y)$ from $D$, the algorithm can generate $k$ samples: $(T_1(x), y), …, (T_k(x), y)$ from $T_1(D), …, T_k(D)$. And, we only count the number of samples drawn from $D$.
>
> In contrast, in multi-distribution learning, the learning algorithm has access to samples from $k$ distributions $D_1, …, D_k$ and it **does not know** the relationship between $D_1, …, D_k$. And, what is counted is the number of samples drawn in total from all $k$ distributions. While, as you noted and based on line 41, it is possible in principle to find a distribution $D$ and $k$ transformations $T_1, …, T_k$ such that $D_i = T_i(D)$. But, that would require the algorithm to **learn** the transformations $T_1, …, T_k$ based on samples from $D_1, …, D_k$ which is going to be more costly than solving the original multi-distribution problem directly.
>
> In our model, we only consider a covariate shift setting, i.e., the labels do not change under transformations (see Line 29). While multi-distribution learning is more general and allows different labeling functions across the $k$ distributions.
>
> Thus, given the above, we can not in general reduce a multi-distribution learning problem instance to a problem instance in our model.
>
> > As described above under weaknesses, why do the lower bounds for multi-distribution learning not apply to the results of Theorem 3.1? Have I misunderstood something about the model in this paper, or perhaps the lower bound instances from [1]?
>
> The lower bounds from [1] do not apply to Theorem 3.1, because in the setting of this theorem, the learning algorithm is provided access to transformations $T_1, …,  T_k$. And, given a dataset $S$ of $m$ samples from $D$ “source”, the learner can generate $S_1 = T_1(S), …, S_k = T_k(S)$ a total of $mk$ samples by applying the transformations $T_1, …, T_k$. We only count the number of samples drawn from $D$ which is $m$.
>
> While the lower bound instance from [1] (e.g. Theorem 4.2) involves $k$ distributions $D_1, …, D_k$ with **disjoint supports** and the learner **does not know** the transformations $T_1, …, T_k$ nor a source $D$ satisfying $T_i(D) = D_i$. The cost of learning such transformations $T_1, …, T_k$ based solely on samples from $D_1, …, D_k$ will be at least $k$ (i.e., drawing at least one sample from each distribution). In general, learning transformations is more costly (and likely an overkill) to achieve the goal of multi-distribution learning.
>
> > Furthermore, as pointed out on line 41, if the domain of the learning problem is
>  then Brenier's Theorem implies that for any instance of a multi-distribution learning problem
>  with sufficiently "nice" distributions, there is a distribution
>  and set of
>  transformation
>  such that
> . Thus, in the case of finite
>  on domain
>  these two models are equivalent so long as only "nice" distributions are considered. The setting of finite
>  is particularly important, as it is the setting for which the main results in this paper give oracle-efficient algorithms (given an ERM oracle for
> ).
>
> As alluded to above, another challenge here is: because our model only considers covariate shifts (i.e., the labeling doesn't change across transformations), in order to reduce multi-distribution learning to our model, the transformations that need to be learned have to be "label preserving". This is another challenge that will likely make it even more difficult to reduce a problem instance of multi-distribution learning to our model.
>
> We hope that this clarifies the differences between multi-distribution learning and our model. We are happy to answer any further questions, and would appreciate an updated response from you.

---

> > ### Comment · Reviewer_RtrA · 2024-08-09
> >
> > Thank you for your response, it definitely clarifies the relationship to multi-distribution learning, and confirms that everything is fine with Theorem 3.2. To make sure I understand correctly: the main difference is that the transformations $T_i$ are given to the learning algorithm in advance, and the learner only pays for one sample from $D$, from which it is able to generate $k$ transformed samples from $T_1(D),\dots T_k(D)$. In contrast, in multi-distribution learning the learner pays for $k$ samples in order to see one sample from each of $D_1,\dots ,D_k$.
> >
> > Assuming the above description is correct, I find the comparison made to multi-distribution learning on line 313 somewhat confusing. While it is true that your algorithms require sample-complexity logarithmic in $k$ rather than linear in $k$, this is entirely because you get $k$ samples for the price of one when compared to multi-distribution learning. As it is currently written, the paragraph starting on line 313 sounds to me like it is claiming an improvement over multi-distribution learning in the finite $k$ case. However, as far as I can tell, in the case of finite $k$ your algorithm is almost identical to the known algorithms for multi-distribution learning, and the reason for the lower sample-complexity is just from the additional assumption that the transformations are known in advance.
> >
> > I would appreciate it if you could comment on whether my above understanding is correct. If so, I think the comparison to multi-distribution learning likely should be re-written, and would also appreciate if the authors could provide details on how they would rewrite it.

---

> ### Author Response · Authors · 2024-08-09
> **Response to Reviewer RtrA**
>
> Thank you for the follow-up response. We address your comments below.
>
> >Thank you for your response, it definitely clarifies the relationship to multi-distribution learning, and confirms that everything is fine with Theorem 3.2. To make sure I understand correctly: the main difference is that the transformations $T_i$ are given to the learning algorithm in advance, and the learner only pays for one sample from $D$, from which it is able to generate $k$ transformed samples from $T_1(D), \dots, T_k(D)$. In contrast, in multi-distribution learning the learner pays for $k$ samples in order to see one sample from each of $D_1,\dots, D_k$.
>
> Yes, we want to confirm that your understanding above is correct. We totally agree that this distinction is crucial and we will revise the comparison on line 313 to clarify that the reduced sample complexity is due to the additional structure of knowing the $k$ transformations, rather than an inherent improvement over multi-distribution learning. Specifically, we plan to include the following revision:
>
> > **Multi-Distribution Learning.** This line of work focuses on the setting where there are $k$ arbitrary distributions $D_1, \dots, D_k$ to be learned uniformly well, where sample access to each distribution $D_i$ is provided [see e.g., 26]. In contrast, our setting involves access to a single distribution $D$ and transformations $\{T_1, \dots, T_k\}$, that together describe the target distributions: $\{T_1(D), \dots, T_k(D)\}$. From a sample complexity standpoint, multi-distribution learning requires sample complexity scaling linearly in $k$ while in our case it is possible to learn with sample complexity scaling logarithmically in $k$ (see Theorem 3.1 and Remark 3.2). The lower sample complexity in our approach is *primarily* due to the assumption that the transformations $\{T_1, \dots, T_k\}$ are known in advance, allowing the learner to generate $k$ samples from a single draw of $D$. In contrast, in multi-distribution learning, the learner pays for $k$ samples in order to see one sample from each of $D_1,\dots, D_k$. Therefore, while the sample complexity is lower in our setting, this advantage arises from the additional information/structure provided rather than an inherent improvement over the more general setting of multi-distribution learning. From an algorithmic standpoint, our reduction algorithms employ similar techniques based on regret minimization and solving zero-sum games [24].
>
> Thank you again for your insightful feedback, which has significantly improved the clarity and accuracy of our paper.

---

> > ### Comment · Reviewer_RtrA · 2024-08-13
> >
> > Thanks for providing the update to the description of multi-distribution RL. I think this makes it much easier for the reader to position the paper relative to related work. I do appreciate the simplicity with which the new model introduced here covers previously studied settings. At the same time I do not entirely see what new insights it provides beyond the specific algorithms for the most important examples from previously studied settings. Overall, based on the discussion with the authors, I will increase my score to 4.

---

### Official Review · Reviewer_JZtQ · 2024-07-13

**Soundness:** 3
**Presentation:** 2
**Contribution:** 2
**Rating:** 4
**Confidence:** 2

**Summary:**

The authors investigate transform-invariant binary classification problems in which the learner observes transformed features during the prediction phase and aims to construct a classifier robust against test-time transformations. Their first contribution is the derivation of a generalization error bound on the worst-case risk of transform-invariant classification, characterized by the VC dimension of the composite class. Subsequently, they propose an algorithm that achieves this generalization bound without direct access to the hypothesis class, relying solely on the empirical risk minimization (ERM) oracle associated with the hypothesis class. The second contribution is the establishment of a generalization error bound on the number of high-risk transformations, which is also characterized by the VC dimension of the composite class. The third result presents a generalization error bound on the worst-case regret, similarly characterized by the composite class's VC dimension. Empirical evaluations were conducted to demonstrate the behavior of the transform-invariant learning algorithm.

**Strengths:**

The paper is well-written and easy to follow. It addresses a crucial topic for machine learning: robustness against transformations. This focus on developing classifiers that remain effective under various input transformations is highly relevant to real-world applications and aligns well with the conference themes.

The authors provide generalization error bounds for various accuracy measures, each corresponding to potential practical scenarios. This comprehensive approach enhances the applicability of their work.

The methodology to minimize worst-case accuracy solely relying on the ERM oracle is innovative. While it adapts the Multiplicative Weights method to the transform-invariant classification problem, the application of this method to this specific domain represents a novel contribution.

**Weaknesses:**

The theoretical results appear to be direct applications of existing uniform bounds on empirical processes and VC dimension theory. The introduction of transform-invariance does not significantly impede the application of these uniform bounds, as they represent worst-case bounds for functions to be optimized. Specifically, Theorem 2.1 is derived by applying the original VC theory (by Vapnik and Chervonenkis) to the composite class. Theorem 4.1 is a straightforward consequence of the optimistic rate by Cortes et al. for the composite class. Theorem 5.1 is also obtained by directly applying the original VC theory to the composite class. Consequently, the technical novelty underlying the theoretical results is not clearly evident.

The authors do not sufficiently elucidate the implications of their experimental results. The significance and benefits of the empirical evaluation are not adequately explained, leaving the reader unclear about the practical impact of the proposed approach.

**Questions:**

- Can you provide more insight into the practical implications of your empirical evaluations? What specific benefits does your transform-invariant learning algorithm offer in real-world scenarios?

**Limitations:**

The authors adequately address the limitations and potential impacts.

---

> ### Author Rebuttal · Authors · 2024-08-04
>
> We thank you for your detailed review. We address your comments below.
>
> We emphasize that our contributions are conceptual and theoretical. At the conceptual-level, to the best of our knowledge, this is the first work to model the OOD/distribution shift problem through a transformation function class, which is relevant to many real-world scenarios as discussed in the introduction. As is well known, studying OOD/distribution shift problems requires specific modeling of the relationship between source and target distributions (e.g., bounded density ratios between source and target distributions, which often don't hold in real-world scenarios). Instead, we propose a new way of modeling the problem through the lens of transformations. This allows us to obtain new learning guarantees against classes of distribution shifts, and while these may not require new sophisticated tools, they nonetheless provide bounds that were not covered by prior work.
>
> > The theoretical results appear to be direct applications of existing uniform bounds on empirical processes and VC dimension theory. The introduction of transform-invariance does not significantly impede the application of these uniform bounds, as they represent worst-case bounds for functions to be optimized. Specifically, Theorem 2.1 is derived by applying the original VC theory (by Vapnik and Chervonenkis) to the composite class. Theorem 4.1 is a straightforward consequence of the optimistic rate by Cortes et al. for the composite class. Theorem 5.1 is also obtained by directly applying the original VC theory to the composite class. Consequently, the technical novelty underlying the theoretical results is not clearly evident.
>
> We believe that it is an asset to be able to leverage the structure of the composition of hypotheses and transformations, and to obtain sample complexity bounds by utilizing existing VC theory (as also highlighted by reviewers RtrA and fSmB), to obtain new insights for this relevant OOD setting.  We further demonstrate this by providing in Section 2.1 several examples highlighting how a simple bound can imply several interesting results and guarantees that are not known before.
>
> > The authors do not sufficiently elucidate the implications of their experimental results. The significance and benefits of the empirical evaluation are not adequately explained, leaving the reader unclear about the practical impact of the proposed approach.
>
> We will add this elaboration to the paper. Please see below for further clarification.
>
> > Can you provide more insight into the practical implications of your empirical evaluations? What specific benefits does your transform-invariant learning algorithm offer in real-world scenarios?
>
> We only provide a simple illustrative experiment showing how the use of transformations can help learn with fewer samples / training data. Specifically, in Figure 1, both learning tasks can be learned better (i.e., higher accuracy) with the use of transformations compared with the baseline of standard training (with no transformations).
>
> Investigating implications of our theoretical results in real-world scenarios is of course an interesting and important future direction, but it lies beyond the scope of this work.

---

> > ### Comment · Reviewer_JZtQ · 2024-08-14
> >
> > Thank you for your response. I prefer to keep the current score, as the results are a consequence of combining the existing results.

---

### Official Review · Reviewer_fcsB · 2024-07-15

**Soundness:** 4
**Presentation:** 4
**Contribution:** 2
**Rating:** 6
**Confidence:** 4

**Summary:**

The work focuses on theoretical aspects of distribution shift, considering various frameworks that expand the standard PAC model of learning.
The paper considers a hypothesis class $\mathcal{H}$ and a class of transformations of the domain $\mathcal{T}$. The learning algorithm is given some data drawn i.i.d. from some distribution $D$ and the overall goal is to produce a classifier that does well not only on fresh samples from the source distribution $D$ but also when a transformation $T$ from $\mathcal{T}$ is applied to $D$.

Overall, the makes assumptions on the VC dimension of the class of functions obtained from composition of $\mathcal{H}$ and $\mathcal{T}$. In some parts of the paper it is further assumed that $\mathcal{T}$ is finite. Various parts of the paper consider different loss functions that depend on the data distribution and the class of transformations $\mathcal{T}$. The goal is to find a hypothesis $\hat{h}$ for which the value of a loss function is not more by $\epsilon$ greater than the best (i.e. smallest) value of the loss function among all functions in the hypothesis class $\mathcal{H}$. (Note that this optimum value also depends on the choice of the loss function.)

The paper considers a number of specific goals in this broad setup:
- In Section 2 the loss of a classifier $\hat{h}$ equals to the worst-case population risk that $\hat{h}$ will have as a result of applying a transformation $T$ from $\mathcal{T}$ to the data distribution. Note that this loss function quantifies the worst-case impact that a transformation from $\mathcal{T}$ can have on the population loss. Section 2 considers the classifier $\hat{h}$ that has the smallest
 introduces and analyzes the classifier obtained by minimizing the empirical worst-case risk that a hypothesis will have as a result of applying a transformation $T$ from $\mathcal{T}$ to the training dataset.
- In Section 3 the class of transformations $\mathcal{T}$ is assumed to be finite and the algorithm has access to an ERM oracle for the hypothesis class $\mathcal{H}$. First, the realizable setting is considered (i.e. the setting in which there is some function in $\mathcal{H}$ that perfectly desribes the data even if it is transformed by an arbitrary transformation in $\mathcal{T}$). The paper shows that the classifier shown to exist in Section(2) can be obtained by (i) augmenting the training dataset by applying various transformations in $\mathcal{T}$ and (ii) applying the ERM oracle to the resulting dataset.
- In Section 3 the paper also considers the agnostic setting. In this setting there can potentially be  a transformation in $\mathcal{T}$ that causes the empirical risk to be non-zero. The paper gives an efficient algorithm that combines calls to the ERM oracle with the multiplicative weight update method to obtain a classifier shown to exist in section 2. Note that the run-time of the algorithm does not include the time to actually minimize the empirical risk when the ERM oracle is called, which can potentially be slow.
- Section 4 considers the case in which there is potentially no classifier whose error stays small when **any** transformation in $\mathcal{T}$ is applied. Instead, Section 4 considers a loss function equal to the **number** of transformations in $\mathcal{T}$ that make the population loss of the classifier greater then epsilon. The paper gives a statistical generalization bound with respect to this loss.
- Section 5 again considers a different loss function. The loss of a hypothesis $\mathcal{h}$ is defined as the worst-case difference (over transformation $T$ in $\mathcal{T}$) between (i) the population risk of $\mathcal{h}$ after transformation $T$ is applied (ii) the lowest population risk among all hypotheses in $\mathcal{H}$. In other words, if there is some transformation $T$ makes the error of $\mathcal{h}$ large, we do not penalize the algorithm if the worst-case error in the hypothesis class  $\mathcal{H}$ is likewise large. With respect to this loss function as well, the paper again gives generalization bounds (similar to Sections 2 and 4) and an algorithmic reduction (Similar to Section 3).

Additionally, an experiment is presented in which a Boolean function is learned from synthetic data, and the function is invariant under a class of permutations.

**Strengths:**

- The question of mitigating distribution shift is central for modern machine learning.
- The paper is written clearly and numerous examples are given.

**Weaknesses:**

- The technical results follow by a fairly direct use of standard generalization bounds, together with standard tools from convex optimization such as multiplicative weights.
- One could argue that one of the most important use cases is when $\mathcal{T}$ form a group of symmetry transformations. As discussed on page 9, in this setting the technique of data augmentation was studied previously. However, the previous work does not enjoy the worst-case error guarantees across transformations.

**Questions:**

No questions.

**Limitations:**

Limitations are discussed adequately. However, I believe it would be helpful to additionally have a dedicated short section titled limitations that briefly summarizes the limitations.

---

> ### Author Rebuttal · Authors · 2024-08-04
>
> Thank you for your detailed review. We address one your comments below.
>
> > One could argue that one of the most important use cases is when $\mathcal{T}$ form a group of symmetry transformations. As discussed on page 9, in this setting the technique of data augmentation was studied previously.
>
> As mentioned in page 9 (Lines 333–338), we emphasize that the technique of data augmentation does not enjoy the worst-case error guarantees across transformations as we prove in this work (using a different learning rule).
>
> > Limitations are discussed adequately. However, I believe it would be helpful to additionally have a dedicated short section titled limitations that briefly summarizes the limitations.
>
> As suggested, we will reiterate the limitations together in a single short section and discuss future open questions.

---

> ### Comment · Reviewer_fcsB · 2024-08-07
>
> Thank you for your response. I updated the text of my review accordingly and I will finalize my rating once the rebuttal discussion with other reviewers concludes. For now, my rating remains 6: Weak Accept.

---

### Official Review · Reviewer_Whvi · 2024-07-23

**Soundness:** 2
**Presentation:** 3
**Contribution:** 3
**Rating:** 4
**Confidence:** 3

**Summary:**

This paper proposed a transformation-based invariant learning framework for OOD generalization. The authors initiated a theoretical study for this framework, investigating learning scenarios where the target class of transformations is either known or unknown. Upper bounds on the sample complexity in terms of the VC dimension of the class composing predictors with transformations were obtained.

**Strengths:**

Nice theoretical results, with various scenarios considered.

**Weaknesses:**

When proposing a new learning rule, it is better to demonstrate its utility by comparing it to existing and similar ones, such as ERM (at least) and DRO (similar).

As the authors mentioned in Related Work, IRM is another popular learning framework for OOD generalization, but some works concluded that ERM may outperform IRM in some cases. Some other works showed that ERM may outperform DRO. Will the learning framework proposed in this paper be underperformed by ERM under some scenarios?

Consider a more realistic definition of $\mathcal{T}$: T(X)=X+R(X), as in the ResNet, because Q and P should not be too far away. In such a case, what will the learning rule, the theoretical guarantee, and the algorithm be like?

What's the connection between this work and the importance weighting for distributional shift (such as [1])?

[1] T. Fang, etc. Rethinking Importance Weighting for Deep Learning under Distribution Shift. NeurIPS 2020.

**Questions:**

Please see the questions mentioned above.

**Limitations:**

N.A.

---

> ### Author Rebuttal · Authors · 2024-08-04
>
> Thank you for your detailed review. We address your comments below.
>
> > When proposing a new learning rule, it is better to demonstrate its utility by comparing it to existing and similar ones, such as ERM (at least) and DRO (similar).
>
> ERM can be viewed as concerned only with the identity transformation map $T(x) = x$. Therefore, when interested in a collection of transformations e.g. rotations, then ERM can be provably shown to do worse than our proposed learning rule in Equation 2. Here is an **Example/Scenario** that further illustrates this difference: when the training set consists of images of upright cats, ERM might learn the feature of "upright," leading the model to fail when the test set is composed of upside-down cat images. Our method, however, would still work in this scenario.
>
> We discuss the relationship with the DRO line of work in (Lines 306–312).
>
> > As the authors mentioned in Related Work, IRM is another popular learning framework for OOD generalization, but some works concluded that ERM may outperform IRM in some cases. Some other works showed that ERM may outperform DRO. Will the learning framework proposed in this paper be underperformed by ERM under some scenarios?
>
> From a theoretical standpoint, ERM and our proposed learning rule (e.g., Equation 2) enjoy different guarantees. When we are only interested in performing well on a single source distribution, then ERM suffices. But, when interested in a collection of distributions parametrized by transformations  $\mathcal{T}$, our learning rule outperforms ERM.
>
> > Consider a more realistic definition of $\mathcal{T}$: $T(x)=x+R(x)$, as in the ResNet, because Q and P should not be too far away. In such a case, what will the learning rule, the theoretical guarantee, and the algorithm be like?
>
> In this case, the learning rule/algorithm corresponds to solving an alternating minimization-maximization problem. The minimization is with respect to parameters of the model responsible for classification (e.g. an MLP on top of the ResNet architecture) and the maximization is with respect to transformations (e.g. the parameters of the $R$ function you describe). The theoretical guarantee is that with enough samples (scaling with VC dimension of ResNet), the learned model will minimize worst-case error (across all $R$ functions).
>
> Also, please see Lines 144–155 for discussion of a similar example concerned with feed-forward neural networks.
>
> > What's the connection between this work and the importance weighting for distributional shift (such as [1])?
>
> Importance weighting addresses the problem of covariate shift / distribution shift from a different angle relying on *bounded* density ratios between source and target distributions. Our work instead considers transformations, i.e., when $P$ and $Q$ are related by a transport map such that $Q = T(P)$. This allows us to address classes of distribution shifts that may be difficult to address with importance weighting, specifically when the density ratio is not bounded. I.e., when distributions $P$ (source) and $Q$ (target) have disjoint supports, the density ratio is not bounded, and therefore we can not rely on importance weighting. However, it may still be possible to learn using the language of transformations that we contribute in this work.

---

### Author Rebuttal · Authors · 2024-08-06

We thank all the reviewers for their helpful input and feedback.

We emphasize that our contributions are conceptual and theoretical. At the conceptual-level, to the best of our knowledge, this is the first work to model the OOD/distribution shift problem through a transformation function class, which is relevant to many real-world scenarios as discussed in the introduction. As is well known, studying OOD/distribution shift problems requires specific modeling of the relationship between source and target distributions (e.g., bounded density ratios between source and target distributions, which often don't hold in real-world scenarios). Instead, we propose a new way of modeling the problem through the lens of transformations. This allows us to obtain new learning guarantees against classes of distribution shifts, and while these may not require new sophisticated tools, they nonetheless provide bounds that were not covered by prior work.

We believe that it is an asset to be able to leverage the structure of the composition of hypotheses and transformations, and to obtain sample complexity bounds by utilizing existing VC theory (as also highlighted by reviewers RtrA and fSmB), to obtain new insights for this relevant OOD setting.  We further demonstrate this by providing in Section 2.1 several examples highlighting how a simple bound can imply several interesting results and guarantees that are not known before.

---

### Decision · Program_Chairs · 2024-09-25

**Decision:**

Accept (poster)

**Comment:**

The authors introduce a new framework for studying transformation-invariant learning and OOD generalization. In this model, a class of invariances (i.e. transformations) are given to the learner. Therefore, each data point can be regarded as multiple data points across domains (sharing the same label). The model is closely related to multi-distribution learning and DRO. In the realizable setting, it is closely related to learning with the data augmentation, and learning under adversarial perturbations.

Overall, the new setting and the theoretical results are interesting. The proof techniques are straightforward (e.g., standard use of uniform convergence). Some of the results are somewhat weak (e.g., the agnostic result only works for finite class of transformations, or the VC bound is tight only for some hypothesis classes). As such, this is a borderline paper.

However, I think the new setting may be of interest to researchers in the field and the shortcomings can be addressed in follow up work. Therefore, I recommend acceptance.